# Fermionic defects of topological phases and logical gates

**Ryohei Kobayashi**

Condensed Matter Theory Center and Joint Quantum Institute, Department of Physics,
University of Maryland, College Park, Maryland 20472 USA

## Abstract

We discuss the codimension-1 defects of (2+1)D bosonic topological phases, where the defects can support fermionic degrees of freedom. We refer to such defects as fermionic defects, and introduce a certain subclass of invertible fermionic defects called "gauged Gu-Wen SPT defects" that can shift self-statistics of anyons. We derive a canonical form of a general fermionic invertible defect, in terms of the fusion of a gauged Gu-Wen SPT defect and a bosonic invertible defect decoupled from fermions on the defect. We then derive the fusion rule of generic invertible fermionic defects. The gauged Gu-Wen SPT defects give rise to interesting logical gates of stabilizer codes in the presence of additional ancilla fermions. For example, we find a realization of the CZ logical gate on the (2+1)D $\mathbb{Z}_2$ toric code stacked with a (2+1)D ancilla trivial atomic insulator. We also investigate a gapped fermionic interface between (2+1)D bosonic topological phases realized on the boundary of the (3+1)D Walker-Wang model. In that case, the gapped interface can shift the chiral central charge of the (2+1)D phase. Among these fermionic interfaces, we study an interesting example where the (3+1)D phase has a spatial reflection symmetry, and the fermionic interface is supported on a reflection plane that interpolates a (2+1)D surface topological order and its orientation-reversal. We construct a (3+1)D exactly solvable Hamiltonian realizing this setup, and find that the model generates the $\mathbb{Z}_8$ classification of the (3+1)D invertible phase with spatial reflection symmetry and fermion parity on the reflection plane. We make contact with an effective field theory, known in literature as the exotic invertible phase with spacetime higher-group symmetry.

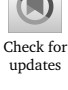

# 1   Introduction

Topologically ordered phases of matter are characterized in general by properties of fusion and braiding of topological defects (excitations). In (2+1) spacetime dimensions, the line operators of anyon excitations are regarded as codimension-2 topological defects of the theory, and their universal properties are described by modular tensor category [1–8]. Also, (2+1)D topological phases in general have codimension-1 defects with non-trivial topological properties, and they all generate the emergent global symmetry of the phases [9–18]. For instance, when the codimension-1 defect is non-invertible, certain anyons near a line defect can be annihilated or created by a local operator. When the codimension-1 defect is invertible, an anyon cannot be annihilated on the defect but rather converted to a topologically distinct anyon upon crossing the defect. In general, these line defects can be thought of as topologically distinct

classes of gapped interfaces between the given topological order and itself. The combined algebraic structure of codimension-1 and codimension-2 defects in (2+1)D topological phases is algebraically described by a unitary fusion 2-category [19,20]. In particular, in the case where codimension-1 defect is invertible and forms a group $G$ under fusion, the algebraic properties of the defects are developed in the study of symmetry-enriched topological phases in terms of $G$-crossed braided tensor categories [21–23].

In general, the symmetry generators of finite Abelian gauge theory are realized as the logical gates on the ground state Hilbert space of a stabilizer code [24–29], where the logical gate is equivalent to sweeping topological defects inserted in the 2d space across the system. Therefore, a systematic understanding of emergent symmetry and the corresponding topological defects is important for the classification of fault-tolerant logical gates in quantum codes as well as the classification of topologically ordered phases.

In this paper, we study the codimension-1 defects of (2+1)D topological phases, where the defect supports fermionic degrees of freedom on it. Such a defect is referred to as a fermionic defect, where we allow the (2+1)D phase to be either bosonic or fermionic. When the bulk (2+1)D phase is fermionic, the defects are topological and generally described in the framework of $G$-crossed category for fermionic topological phases [30,31]. When the (2+1)D phase is bosonic, the fermions are introduced only at the codimension-1 defect, and the defect is no longer mobile at the level of lattice models. In general, a fermionic defect of a bosonic topological phase can shift the self-statistics of the anyons. For instance, the invertible fermionic defect realized in the (2+1)D $\mathbb{Z}_2$ toric code permutes the magnetic particle $m$ to a fermionic particle $\psi$. Also, the invertible fermionic defect in the (2+1)D double semion model permutes a semion $s$ to an anti-semion $\bar{s}$, again shifting the spin of the anyon by $1/2$. We construct Pauli stabilizer models of twisted or untwisted $\mathbb{Z}_2$ gauge theory in (2+1)D with an insertion of invertible fermionic defect shifting the spins of the anyons as above.

The lattice model of an invertible fermionic defect for $\mathbb{Z}_2$ gauge theory is constructed in the following fashion. We start with a (2+1)D SPT phase with $\mathbb{Z}_2$ symmetry, with a location of the (1+1)D $\mathbb{Z}_2 \times \mathbb{Z}_2^f$ Gu-Wen SPT phase [32] on the codimension-1 defect. Note that we introduce additional fermions on the codimension-1 submanifold to define a fermionic defect on it, so the $\mathbb{Z}_2^f$ symmetry is only defined at the defect. We then gauge the $\mathbb{Z}_2$ symmetry of the whole system, and we obtain the $\mathbb{Z}_2$ gauge theory with an insertion of invertible fermionic defect. Similar constructions of the bosonic defects based on the insertion of lower dimensional SPT phase are found in [24–26,33].

After introducing the lattice models, we then discuss the general invertible fermionic defect realized in (2+1)D bosonic topological quantum field theory (TQFT). We introduce a specific class of invertible fermionic defects referred to as "gauged Gu-Wen SPT defects", which is a certain generalization of the fermionic defects in the above lattice models obtained from (1+1)D $\mathbb{Z}_2 \times \mathbb{Z}_2^f$ Gu-Wen SPT phase. The gauged Gu-Wen SPT defect is denoted as $U_{\xi,a}$, where $\xi$ denotes the spin structure of the defect, and $a$ is Abelian bosonic particle of TQFT with $\mathbb{Z}_2$ fusion rule, i.e., $\theta_a = 1, a \times a = 1$. In general, the gauged Gu-Wen SPT defect is expressed in terms of a condensation defect of the particle $a$ [34], which is obtained by gauging the $\mathbb{Z}_2$ symmetry generated by the Wilson line of $a$, where the gauging is performed only at the defect. Such a process of gauging the symmetry restricted to the codimension-1 defect is referred to as 1-gauging in [34]. One can show that the gauged Gu-Wen SPT defect always gives an invertible defect with $\mathbb{Z}_2$ fusion rule, and causes a permutation of anyons shifting their self-statistics.

The benefit of introducing the notion of gauged Gu-Wen SPT defect is that one can obtain a canonical form of the general invertible fermionic defect realized in (2+1)D bosonic TQFT. Concretely, we show that any invertible fermionic defect of (2+1)D bosonic TQFT is expressed in the form of $U_{\xi,a} \times V$, where $U_{\xi,a}$ is a gauged Gu-Wen defect with some choice of the anyon $a$, and $V$ is a bosonic invertible defect that induces an automorphism of the TQFT [22]. Based

on this canonical form, we derive the fusion rule of general fermionic invertible defects.

After discussing the generality of the invertible fermionic defect of (2+1)D bosonic TQFT, we investigate the application of the fermionic defects to the logical gates of the Pauli stabilizer models. Roughly speaking, when we introduce the fermions in the whole space instead of being restricted to the defect, one can sweep the fermionic defect over the whole space, which acts on the state by emergent global symmetry that corresponds to the defect. At the level of the lattice models, such an action of the emergent symmetry on the state should be realized as the logical gate acting on the code space of the stabilizer model.

Associated with the fermionic defects realized in the (2+1)D $\mathbb{Z}_2$ toric code, we obtain a new non-Pauli Clifford logical gate for the (2+1)D $\mathbb{Z}_2$ toric code stacked with a (2+1)D ancilla trivial atomic insulator. This logical gate realizes the $CZ$ gate in the code space of the $\mathbb{Z}_2$ toric code, and implements the permutation of anyons $m \leftrightarrow \psi$. We emphasize that this logical gate shifting the self-statistics of the anyons is made possible by introducing ancilla fermionic degrees of freedom. We also construct a logical gate for the Pauli stabilizer model of the double semion theory stacked with an atomic insulator. The logical gate implements the SWAP gate in the code space, which corresponds to exchange of anyons $s \leftrightarrow \bar{s}$.

Next, we discuss the fermionic gapped interface of the (2+1)D bosonic topological phases, which is realized on the boundary of (3+1)D Walker-Wang model [35]. Here, an interface means interpolation of two theories which are not necessarily identical, by a codimension-1 domain wall. Here, the fermionic interface of (2+1)D phase is realized as a termination of the fermionic interface in the (3+1)D bulk. In this setup, one can realize an "anomalous" gapped interface of (2+1)D TQFTs which cannot be realized in standalone (2+1)D phases not coupled with the bulk. Concretely, such a fermionic interface realized on the boundary can shift the chiral central charge of the (2+1)D phase.

In this paper, we consider an interesting example of such an anomalous interface, where the (3+1)D phase has a spatial reflection symmetry and the fermionic interface is supported at the 2d reflection plane. The interface interpolates between (3+1)D Walker-Wang model based on $U(1)_2$ topological order and its orientation-reversal. On the boundary, we have the fermionic interface between $U(1)_2$ and $U(1)_{-2}$ TQFTs at the reflection plane, which shifts the chiral central charge by $-2$. We construct an exactly solvable lattice Hamiltonian in (3+1)D that realizes this fermionic interface. This model turns out to give a (3+1)D invertible topological phase with the spatial reflection symmetry, together with $\mathbb{Z}_2^f$ symmetry localized at the reflection plane. We show that our (3+1)D lattice model generates the $\mathbb{Z}_8$ classification of the invertible phase with the above combination of global symmetries.

Our (3+1)D lattice model is effectively described by an invertible TQFT which is called exotic invertible phase in [36,37]. While the $\mathbb{Z}_2^f$ fermion parity in the whole spacetime corresponds to the spin structure of spacetime manifold at the level of effective field theory, the $\mathbb{Z}_2^f$ symmetry localized at the reflection plane is regarded as a certain spacetime structure which is a 2-group describing the mixture of $\mathbb{Z}_2$ 1-form symmetry and spacetime Lorentz symmetry. We explain the relation between our lattice model and the effective field theory, and describe the $\mathbb{Z}_8$ classification from field theoretical perspective.

This paper is organized as follows. We start with construction of lattice models for fermionic invertible defects of (2+1)D $\mathbb{Z}_2$ gauge theory in Sec. 2.1. We then introduce a gauged Gu-Wen SPT defects in terms of condensation defects in Sec. 2.2, and derive a canonical form of the invertible fermionic defect and fusion rules in the rest of Sec. 2. In Sec. 3, we construct the logical gates of stabilizer codes stacked with atomic insulator, including the $CZ$ gate of the $\mathbb{Z}_2$ toric code. In Sec. 4, we study the fermionic interface between (3+1)D Walker-Wang model, and discuss the connection to exotic invertible phase that follows $\mathbb{Z}_8$ classification. Review of concepts used in this paper and detailed calculations are relegated to appendices.

# 2 Fermionic defects of (2+1)D bosonic topological phases

In this section, we consider the codimension-1 fermionic defect of the (2+1)D bosonic topological phases, mainly focusing on the invertible defects. After introducing simple lattice models of the fermionic defects, we describe a generic fermionic invertible defect of (2+1)D bosonic TQFT. We derive a canonical form of the fermionic invertible defect in terms of a specific condensation defect called a gauged Gu-Wen SPT defect, and discuss the fusion rule of fermionic invertible defects.

## 2.1 Lattice models for fermionic defects

Here we discuss examples of the lattice models of (2+1)D bosonic topological phases that host the fermionic defects. We obtain the invertible fermionic defect in the Pauli stabilizer model of $\mathbb{Z}_2$ toric code and the double semion model, both of which shift the self-statistics of the anyons.

### 2.1.1 Review: Gauging $\mathbb{Z}_2$ symmetry on lattice

We first review the procedure of gauging 0-form $\mathbb{Z}_2$ symmetry in (2+1)D lattice systems following [25], which is used to construct the lattice models of $\mathbb{Z}_2$ gauge theory in the presence of defects.

Let us consider a 2d square lattice where the Hilbert space is formed by qubits at vertices. We start with a Hamiltonian respecting a global $\mathbb{Z}_2$ symmetry $\prod_v X_v$. The terms in the Hamiltonian are generated by $X_v$ and $Z_v Z_{v'}$, where $v, v'$ are neighboring vertices in the lattice.

Then, we reformulate the $\mathbb{Z}_2$ symmetric subspace of the full Hilbert space and the symmetric operators in terms of new degrees of freedom, summarized in Fig. 1. Before gauging the $\mathbb{Z}_2$ symmetry (left side of Fig. 1), the symmetric subspace of the full Hilbert space consists of a tensor product of qubits placed at vertices and a global $\mathbb{Z}_2$ symmetry constraint $\prod_v X_v = 1$. After gauging (right side of Fig. 1), the dual Hilbert space has qubits on the edges and local gauge constraints $\prod_{e \subset f} Z_e = 1$ in addition to one-form symmetry constraints $\prod_{e \subset \gamma} Z_e = 1$ for all closed loops $\gamma \in Z_1(M, \mathbb{Z}_2)$ (where $Z_1$ denotes 1-cycles). If we start with the trivial insulator with $\mathbb{Z}_2$ symmetry $H = -\sum_v X_v$, the dual Hamiltonian realizes the $\mathbb{Z}_2$ toric code model after gauging $\mathbb{Z}_2$ symmetry.[1]

### 2.1.2 $\mathbb{Z}_2$ toric code

The simplest case of the fermionic codimension-1 defect in (2+1)D topological phase can be found in the (2+1)D untwisted $\mathbb{Z}_2$ gauge theory (toric code). To obtain the defect, we start with the (2+1)D $\mathbb{Z}_2$ symmetric trivial insulator $H_0 = -\sum_v X_v$, and replace the Hamiltonian on the codimension-1 submanifold with a (1+1)D $\mathbb{Z}_2 \times \mathbb{Z}_2^f$ Gu-Wen fermionic SPT phase [32] (see the left side of Fig. 2). Here, we additionally introduce a complex fermion (a pair of Majorana fermions $\gamma, \gamma'$) on each edge of the defect line to put the (1+1)D fermionic SPT phase on it. We then gauge the $\mathbb{Z}_2$ symmetry and obtain a model for the $\mathbb{Z}_2$ gauge theory with a defect. Before gauging the $\mathbb{Z}_2$ symmetry, the Hamiltonian for the $\mathbb{Z}_2 \times \mathbb{Z}_2^f$ Gu-Wen SPT phase on the 1d defect is given by

$$H_{\text{GW}} = -\sum_j i \gamma_{j-\frac{1}{2}} X_j \gamma'_{j+\frac{1}{2}} - \sum_j i Z_j (\gamma_{j+\frac{1}{2}} \gamma'_{j+\frac{1}{2}}) Z_{j+1}, \tag{1}$$

---

[1] We impose the gauge constraint $\prod_{e \subset f} Z_e = 1$ energetically, which realizes the $Z$-plaquette term in the gauged Hamiltonian.

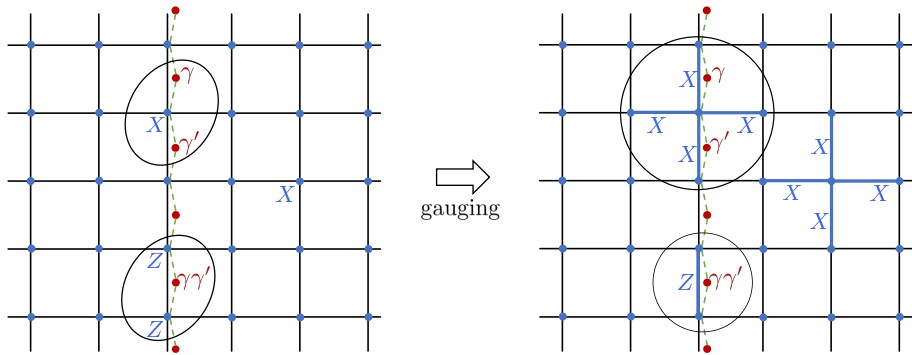

Figure 1: Gauging $\mathbb{Z}_2$ symmetry [25]. Left: each dot represents a qubit on a vertex. We consider the symmetric sector of the Hilbert space: $\prod_v X_v = 1$. Symmetric operators are generated by the single $X_v$ and the product of adjacent $Z_v$. Right: The Hilbert space contains qubits at all edges, with the gauge constraint $\prod_{e \subset f} Z_e = 1$ for each face $f$. For non-simply connected manifolds, there are additional constraints that the product of $Z_e$ along any cycle equals $+1$.

Figure 2: Gauging $\mathbb{Z}_2$ symmetry in the presence of the (1+1)D Gu-Wen SPT phase to obtain a fermionic defect of a $\mathbb{Z}_2$ toric code in (2+1)D.

where we label the vertices on the 1d defect by integer $j \in \mathbb{Z}$. The Hamiltonian after gauging $\mathbb{Z}_2$ symmetry is described in the right side of Fig. 2. The Hamiltonian is given by the standard $\mathbb{Z}_2$ toric code away from the 1d defect, while the star term $-\prod_{v \in e} X_e$ on a vertex $v$ is modified on the 1d defect as $-i\gamma_{j-\frac{1}{2}}\gamma'_{j+\frac{1}{2}}\prod_{v_j \subset e} X_e$, and we have an additional term $-i\gamma_e \gamma'_e Z_e$ on each edge of the defect.

One can see that this 1d defect realizes the permutation of anyons $m \leftrightarrow \psi$, while leaving $e$ invariant. Indeed, the line operator for the $e$ particle (product of $Z$ operators along the line) can freely pass through the defect, so the defect leaves $e$ invariant. Meanwhile, the line operator for the $m$ particle violates the $-i\gamma_e \gamma'_e Z_e$ term on the defect, and the line operator across the defect gets associated with the additional $e$ line as described in Fig. 3, so that the line operator commutes with the Hamiltonian on the defect. The defect hence realizes the permutation $m \to \psi$.

### 2.1.3 Double semion model

Let us consider one more example of a codimension-1 fermionic defect in the (2+1)D double semion model, which realizes the (2+1)D $\mathbb{Z}_2$ gauge theory twisted by a non-trivial element of group cohomology $H^3(B\mathbb{Z}_2, U(1))$. A fermionic defect of the double semion model can also be constructed by a decoration of the Gu-Wen SPT phase on the codimension-1 defect. To

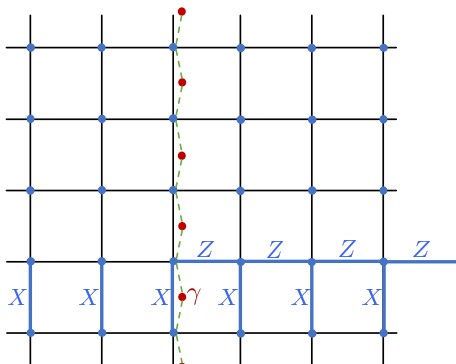

Figure 3: The line operator of a $m$ particle across the defect. One can see that the $e$ particle emanates from the intersection between the $m$ line and the defect, realizing the permutation action of the defect $m \to m \times \psi$.

describe a lattice Hamiltonian model, we follow the construction of the double semion model given in [38], where they obtained a double semion model starting with the $\mathbb{Z}_4$ toric code with anyons $\{m^j e^k\}$ with $j, k \in \mathbb{Z}_4$, and then condensing a boson $m^2 e^2$. After condensing $m^2 e^2$, the resulting topological order is given by a double semion model with a semion $me$ and anti-semion $me^3$.

Following their construction, we firstly consider the $\mathbb{Z}_4$ toric code, in the presence of the 1d defect that induces the permutation of anyons given by

$$m^j e^k \to m^j e^{k+2j}, \quad j, k \in \mathbb{Z}_4. \tag{2}$$

The above defect of the $\mathbb{Z}_4$ toric code is obtained by a decoration of the $\mathbb{Z}_4 \times \mathbb{Z}_2^f$ Gu-Wen SPT phase for the $\mathbb{Z}_4$ symmetric trivial insulator, and then gauging $\mathbb{Z}_4$ symmetry. The process of gauging $\mathbb{Z}_4$ symmetry on the lattice is done by a straightforward generalization of Fig. 1 for gauging $\mathbb{Z}_2$ symmetry. Concretely, we start with a square lattice with a four-dimensional qudit on each vertex, characterized by generalized $\mathbb{Z}_4$ Pauli operators $Z, X$ obeying the $\mathbb{Z}_4$ clock and shift algebra

$$Z^4 = X^4 = 1, \qquad ZX = iXZ. \tag{3}$$

Before gauging, we have a Hamiltonian for a trivial insulator $H = -\sum_v (X_v + X_v^\dagger)$ with $\mathbb{Z}_4$ symmetry $U = \prod_v X_v$. To introduce a defect, we additionally put a complex fermion (a pair of Majorana fermions $\gamma, \gamma'$) on each edge of the 1d defect. We then replace the Hamiltonian on the 1d defect line with the (1+1)D Gu-Wen SPT with $\mathbb{Z}_4 \times \mathbb{Z}_2^f$ symmetry,

$$H_{\text{GW}} = -\left( \sum_j i\gamma_{j-\frac{1}{2}} X_j \gamma'_{j+\frac{1}{2}} + \text{h.c.} \right) - \sum_j iZ_j^2 (\gamma_{j+\frac{1}{2}} \gamma'_{j+\frac{1}{2}}) Z_{j+1}^2. \tag{4}$$

After gauging the $\mathbb{Z}_4$ symmetry, we get a Hamiltonian for the $\mathbb{Z}_4$ toric code with the star term modified as described in the right side of Fig. 4, and also with the term $-i\gamma_e \gamma'_e Z_e^2$ introduced on each edge of the 1d defect. One can see that the defect realizes the permutation of anyons Eq. (2).

Next, we obtain the double semion model by condensing the boson $m^2 e^2$ of the $\mathbb{Z}_4$ toric code in the presence of the defect. The condensation of $e^2 m^2$ is performed on the lattice model by adding terms $C_e$ to the Hamiltonian that correspond to hopping of the anyon $e^2 m^2$. See Fig. 5 for the definition of $C_e$ on a horizontal or vertical edge of the square lattice, which is

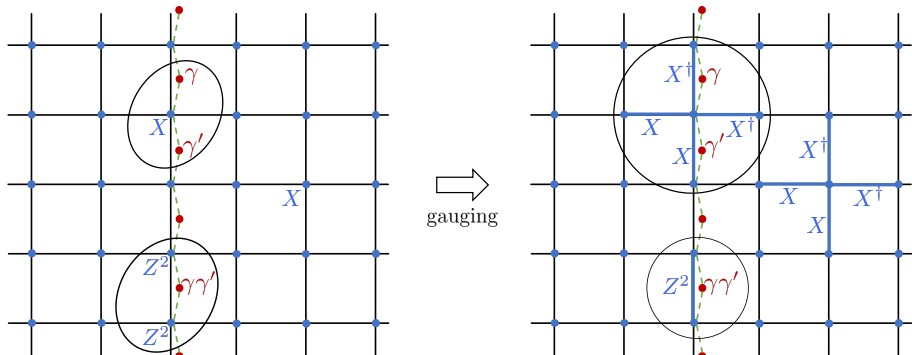

Figure 4: Gauging $\mathbb{Z}_2$ symmetry in the presence of the (1+1)D Gu-Wen SPT phase to obtain a fermionic defect of a $\mathbb{Z}_2$ toric code in (2+1)D.

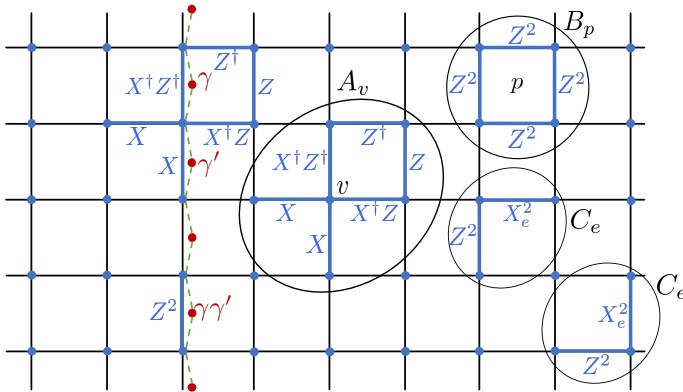

Figure 5: The Hamiltonian for the double semion model in the presence of the fermionic defect.

regarded as a short open Wilson line operator for the anyon $e^2m^2$ along an edge $e$. The Hamiltonian is then constructed by picking up the stabilizers of the $\mathbb{Z}_4$ toric code that commutes with all the hopping terms $\{C_e\}$. That is, we sum over all the generators for the subgroup of the stabilizer group that commutes with $\{C_e\}$. The Hamiltonian after condensation of $e^2m^2$ is then given by

$$H = -\sum_{v\in\text{vertex}} A_v - \sum_{p\in\text{plaquette}} B_p - \sum_{e\in\text{edge}} C_e, \tag{5}$$

which is valid away from the defect, see Fig. 5 for definitions of $A_v, B_p$. The Hamiltonian on the defect is modified and coupled with the fermions at the defect, as shown in Fig. 5. Note that the vertex term $A_v$ is dressed with fermion operators $i\gamma_{j-\frac{1}{2}}\gamma'_{j+\frac{1}{2}}$, and we also have a term $-iZ_e^2\gamma_e\gamma'_e$ on each edge $e$ of the defect.

One can see that the defect obtained after condensing $m^2e^2$ realizes the permutation of anyons $s \to \bar{s}$. This can be seen by considering a line operator of the semion passing through the defect, as shown in Fig. 6. In order to make the line operator commute with the Hamiltonian on the defect, the line operator for $Z^2$ that corresponds to a boson $s\bar{s}$ must emit from the intersection between the line operator for a semion and the defect.

## 2.2 Gauged Gu-Wen SPT defect in (2+1)D bosonic topological phases

In the above discussions, we have seen that the fermionic defect of (2+1)D $\mathbb{Z}_2$ gauge theory can be obtained by the (1+1)D $\mathbb{Z}_2 \times \mathbb{Z}_2^f$ Gu-Wen SPT phase with support on a codimension-1

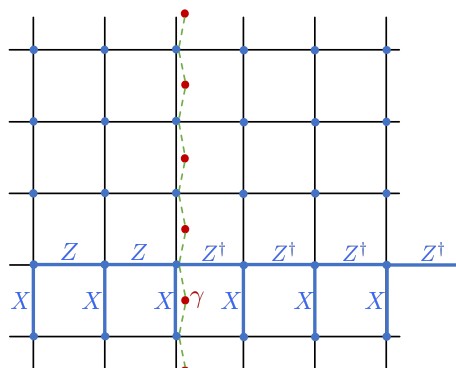

Figure 6: The line operator for a semion passing through the defect.

defect of (2+1)D spacetime. We note that while the $\mathbb{Z}_2$ symmetry of $\mathbb{Z}_2 \times \mathbb{Z}_2^f$ Gu-Wen phase is gauged and identified as $\mathbb{Z}_2$ gauge group in the bulk, the $\mathbb{Z}_2^f$ symmetry is not gauged and only defined on the defect. We refer to such a defect obtained by the (1+1)D $\mathbb{Z}_2 \times \mathbb{Z}_2^f$ Gu-Wen SPT phase as a gauged Gu-Wen SPT defect.

In this subsection, we give the expression of the gauged Gu-Wen SPT defect in terms of a condensation defect [34]. Here, condensation defect means a codimension-1 defect which is obtained by gauging the global symmetry generated by Wilson lines of anyons, where the gauging is performed only on the defect. We demonstrate that the gauged Gu-Wen SPT defect can in general be obtained by gauging the symmetry generated by an Wilson line of a certain Abelian anyon. Using the expression as a condensation defect, we show that the gauged Gu-Wen SPT defect can be defined for a generic (2+1)D bosonic topological phase, if the bosonic topological phase has an Abelian boson that obeys the $\mathbb{Z}_2$ fusion rule.

### 2.2.1 Gauged Gu-Wen SPT defect of $\mathbb{Z}_2$ toric code as a condensation defect

Let us consider the simplest example of the gauged Gu-Wen SPT defect of the (2+1)D $\mathbb{Z}_2$ toric code, where the lattice model was discussed in Sec. 2.1. The (2+1)D $\mathbb{Z}_2$ toric code is effectively described in terms of $\mathbb{Z}_2$ gauge theory with the action

$$\pi \int_{M^3} \delta a \cup b \,, \tag{6}$$

with $a, b \in C^1(M^3, \mathbb{Z}_2)$ the $\mathbb{Z}_2$ gauge fields. The equation of motion for $b$ yields $\delta a = 0$. This theory has the $\mathbb{Z}_2 \times \mathbb{Z}_2$ 1-form symmetry generated by

$$W(\gamma) = \exp\left(i\pi \int_\gamma a\right), \qquad V(\gamma') = \exp\left(i\pi \int_{\gamma'} b\right). \tag{7}$$

Also, the gauged Gu-Wen SPT defect for this theory is described by the partition function of (1+1)D Gu-Wen SPT phase on the 2d defect,

$$U_\xi(\Sigma) = \exp\left(i\pi \int_\Sigma q_\xi(a)\right), \tag{8}$$

where $\Sigma$ is an oriented 2d surface supporting the defect, and $\xi$ is a spin structure of $\Sigma$ that reflects the $\mathbb{Z}_2^f$ symmetry on the defect. The integral of $q_\xi(a)$ is an action for the $\mathbb{Z}_2 \times \mathbb{Z}_2^f$ Gu-Wen SPT phase [39], which is known to be a $\mathbb{Z}_2$-valued quadratic function of $a \in H^1(\Sigma, \mathbb{Z}_2)$,

$$\int_\Sigma q_\xi(a + a') = \int_\Sigma q_\xi(a) + q_\xi(a') + a \cup a' \,. \tag{9}$$

The above defect $U_\xi(\Sigma)$ can be regarded as a condensation defect, by introducing dynamical gauge fields $\Gamma, \Gamma' \in Z^1(\Sigma, \mathbb{Z}_2)$ which are defined on the 2d defect $\Sigma$. One can then rewrite the expression of the defect as

$$
\begin{aligned}
U_\xi(\Sigma) &= \frac{1}{|H^1(\Sigma, \mathbb{Z}_2)|} \sum_{\Gamma, \Gamma' \in H^1(\Sigma, \mathbb{Z}_2)} \exp\left(i\pi \int_\Sigma a \cup \Gamma' + \Gamma \cup \Gamma' + q_\xi(\Gamma)\right) \\
&= \frac{1}{|H^1(\Sigma, \mathbb{Z}_2)|} \sum_{\Gamma, \Gamma' \in H^1(\Sigma, \mathbb{Z}_2)} W(\gamma') \exp\left(i\pi \int_\Sigma q_\xi(\Gamma + \Gamma') + q_\xi(\Gamma')\right) \\
&= \frac{1}{\sqrt{|H^1(\Sigma, \mathbb{Z}_2)|}} \mathrm{Arf}(\xi) \sum_{\Gamma' \in H^1(\Sigma, \mathbb{Z}_2)} W(\gamma') \exp\left(i\pi \int_\Sigma q_\xi(\Gamma')\right),
\end{aligned}
\tag{10}
$$

where $\gamma' \in H_1(\Sigma, \mathbb{Z}_2)$ is the Poincaré dual of $\Gamma' \in H^1(\Sigma, \mathbb{Z}_2)$, and $\mathrm{Arf}(\xi)$ is the Arf invariant of the quadratic form $\int q_\xi(\Gamma)$ defined as

$$
\mathrm{Arf}(\xi) = \frac{1}{\sqrt{|H^1(\Sigma, \mathbb{Z}_2)|}} \sum_{\Gamma \in H^1(\Sigma, \mathbb{Z}_2)} \exp\left(i\pi \int_\Sigma q_\xi(\Gamma)\right),
\tag{11}
$$

which is regarded as a partition function of a Kitaev chain in (1+1)D [40]. The sum over gauge fields $\Gamma'$ in Eq. (10) is regarded as gauging the symmetry generated by $W(\gamma')$ on the defect $\Sigma$, in the presence of the discrete torsion term $\exp\left(i\pi \int q_\xi(\Gamma')\right)$. This gives the expression of the defect $U_\xi(\Sigma)$ as a condensation defect obtained by gauging the symmetry $W(\gamma')$.

### 2.2.2 Gauged Gu-Wen SPT defect for general bosonic topological phase

As a natural generalization, one can also consider the above condensation defect $U_\xi(\Sigma)$ for a general (2+1)D bosonic topological phase. Suppose that the bosonic phase has an Abelian anyon $a$ following the $\mathbb{Z}_2$ fusion rule, $a \times a = 1$. Let us write the Wilson line operator for the anyon $a$ as $W_a(\gamma)$.

If we want to obtain the condensation defect $U_\xi(\Sigma)$ by gauging the symmetry generated by the Wilson line $W_a(\gamma)$ on the defect $\Sigma$, the symmetry $W_a(\gamma)$ restricted to the defect $\Sigma$ must be free of 't Hooft anomaly. The line operator $W_a(\gamma)$ generates a 0-form symmetry on the 2d defect, and the 't Hooft anomaly for this 0-form symmetry is present when the spin of the anyon $a$ is given by $\theta_a = \pm i$, and absent when $\theta_a = \pm 1$ [34]. When the anyon has the spin $\theta_a = \pm 1$, i.e., $a$ is a boson or fermion, one can obtain the condensation defect $U_{\xi,a}(\Sigma)$ for $W_a(\gamma)$ as

$$
U_{\xi,a}(\Sigma) = \frac{1}{\sqrt{|H^1(\Sigma, \mathbb{Z}_2)|}} \mathrm{Arf}(\xi) \sum_{\Gamma \in H^1(\Sigma, \mathbb{Z}_2)} W_a(\gamma) \exp\left(i\pi \int_\Sigma q_\xi(\Gamma)\right),
\tag{12}
$$

where $\gamma \in H_1(\Sigma, \mathbb{Z}_2)$ is the Poincaré dual of $\Gamma$. The fusion rule of the condensation defect $U_\xi(\Sigma)$ depends on the statistics of the anyon $a$, which is summarized as follows.

- When $a$ is a boson, the defect $U_{\xi,a}(\Sigma)$ is invertible and obeys the $\mathbb{Z}_2$ fusion rule, $U_{\xi,a} \times U_{\xi,a} = 1$. The defect $U_{\xi,a}(\Sigma)$ induces the permutation of anyons given by

$$
\begin{cases} p \to p, & \text{if } M_{p,a} = 1, \\ p \to p \times a, & \text{if } M_{p,a} = -1, \end{cases}
\tag{13}
$$

  where $p$ is an arbitrary anyon of the bosonic topological phase, and $M_{p,a} = \pm 1$ is mutual braiding between anyons $p, a$. The derivation for the symmetry action on anyons in

Eq. (13) is given in Appendix A. Note that the symmetry action shifts the spin of the anyon $p$ by 1/2 when $M_{p,a} = -1$, since $\theta_{p \times a} = \theta_p \theta_a M_{p,a} = -\theta_p$. This is regarded as a generalization of the gauged Gu-Wen SPT defect in $\mathbb{Z}_2$ gauge theory considered in Sec. 2.1.

- When $a$ is a fermion, the defect $U_{\xi,a}(\Sigma)$ is non-invertible and obeys a fusion rule

$$U_{\xi,a}(\Sigma) \times U_{\xi,a}(\Sigma) = \frac{1}{\sqrt{|H^1(\Sigma, \mathbb{Z}_2)|}} \mathrm{Arf}(\xi) U_{\xi,a}(\Sigma), \tag{14}$$

$$U_{\xi,a}(\Sigma) \times W_a(\gamma) = \exp\left(i\pi \int_\Sigma q_\xi(\Gamma)\right) U_{\xi,a}(\Sigma), \tag{15}$$

where the second fusion rule is defined by pushing a closed loop $\gamma$ to a surface $\Sigma$, and $\Gamma$ is the Poincaré dual of $\gamma$.[2] The fusion rule Eq. (15) implies that the Wilson line operator for the fermion $a$ gets absorbed by the defect $U_{\xi,a}(\Sigma)$. This means that the fermion $a$ is condensed and can terminate at the defect $\Sigma$. Physically, the defect $U_{\xi,a}(\Sigma)$ corresponds to condensing a composite boson formed by the fermion $a$ and the local fermion at the defect. This defect is an analogue of the Cheshire string [41, 42] for condensation of a fermionic particle.

In this paper, we mainly focus on the case that the anyon $a$ is a boson, where $U_{\xi,a}(\Sigma)$ generates an invertible $\mathbb{Z}_2$ symmetry. We refer to the defect $U_\xi$ in such a setup, i.e, with the Abelian boson $a$ with the $\mathbb{Z}_2$ fusion rule $a \times a = 1$, as the gauged Gu-Wen SPT defect.

### 2.2.3 Bosonization of gauged Gu-Wen SPT defects

The gauged Gu-Wen SPT defect is not quite topological, since it depends on the spin structure of the defect which is not defined globally on the spacetime. One can obtain a topological defect of the (2+1)D bosonic topological phase by gauging $\mathbb{Z}_2^f$ symmetry of the gauged Gu-Wen defect, which yields a bosonic defect independent of the spin structure. These topological defects are given by summing over spin structures of $U_{\xi,a}(\Sigma)$,

$$\widetilde{U}_a(\Sigma) := \frac{1}{|H^1(\Sigma, \mathbb{Z}_2)|} \sum_\xi U_{\xi,a}(\Sigma), \tag{17}$$

where we sum over the $|H^1(\Sigma, \mathbb{Z}_2)|$ distinct spin structures of the defect $\Sigma$. This sum is explicitly performed in Appendix A, and $\widetilde{U}_a(\Sigma)$ can be simply expressed as

$$\widetilde{U}_a(\Sigma) = \frac{1}{\sqrt{|H^1(\Sigma, \mathbb{Z}_2)|}} \sum_{\Gamma \in H^1(\Sigma, \mathbb{Z}_2)} W_a(\gamma), \tag{18}$$

which is the condensation defect obtained by gauging the symmetry generated by the Wilson line operator $W_a(\gamma)$ on the defect $\Sigma$ [34]. The properties of this condensation defect was studied in [34], which is summarized as follows.

---

[2]The above fusion rules are derived by using the fusion rule of line operators which is valid when $a$ is a fermion,

$$W_a(\gamma) W_a(\gamma') = W_a(\gamma + \gamma')(-1)^{\sharp \mathrm{int}(\gamma, \gamma')}, \tag{16}$$

where $\gamma, \gamma'$ are closed loops embedded in $\Sigma$, and $\sharp\mathrm{int}(\gamma, \gamma')$ is the intersection number of $\gamma, \gamma'$ evaluated in $\Sigma$.

- When $a$ is a boson, the defect $\widetilde{U}_a(\Sigma)$ is non-invertible and obeys a fusion rule

$$\widetilde{U}_a(\Sigma) \times \widetilde{U}_a(\Sigma) = \frac{1}{\sqrt{|H^1(\Sigma, \mathbb{Z}_2)|}} \widetilde{U}_a(\Sigma), \tag{19}$$

$$\widetilde{U}_a(\Sigma) \times W_a(\gamma) = \widetilde{U}_a(\Sigma). \tag{20}$$

The defect $\widetilde{U}_a(\Sigma)$ is understood as the Cheshire string obtained by condensing the boson $a$ at the defect.

- When $a$ is a fermion, the defect $\widetilde{U}_a(\Sigma)$ is invertible and generates the $\mathbb{Z}_2$ global symmetry $\widetilde{U}_a \times \widetilde{U}_a = 1$. The $\mathbb{Z}_2$ symmetry generated by $\widetilde{U}_a(\Sigma)$ induces the permutation of anyons given by

$$\begin{cases} p \rightarrow p, & \text{if } M_{p,a} = 1, \\ p \rightarrow p \times a, & \text{if } M_{p,a} = -1, \end{cases} \tag{21}$$

where $p$ is an arbitrary anyon of the bosonic topological phase. Note that this symmetry action preserves the self-statistics of the anyon; $\theta_{p \times a} = \theta_p \theta_a M_{p,a} = \theta_p$ when $M_{p,a} = -1$.

Note that the bosonic defect $\widetilde{U}_a$ becomes non-invertible when $a$ is a boson while invertible when $a$ is a fermion, which is in contrast to the fermionic defect $U_{\xi,a}$. The condensation of an anyon with the physical fermion for $U_{\xi,a}$ gives rise to a completely different fusion rule from the case without the physical fermion.

## 2.3 All fermionic invertible defects are generated by gauged Gu-Wen SPT defects and bosonic invertible defects

In the above discussions, we have seen that the gauged Gu-Wen SPT defect $U_{\xi,a}$ gives an example of invertible fermionic defects of (2+1)D bosonic topological phase. Here, we derive the canonical form of a fermionic invertible symmetry defect in the (2+1)D topological phase using the gauged Gu-Wen SPT defect. That is, we show that all the codimension-1 fermionic invertible symmetry defects are expressed as a fusion $U_{\xi,a}(\Sigma) \times V(\Sigma)$, where $U_{\xi,a}(\Sigma)$ is a gauged Gu-Wen SPT defect for some Abelian $\mathbb{Z}_2$ boson $a$, and $V(\Sigma)$ is some bosonic invertible defect that induces an automorphism of modular tensor category [22].[3]

This allows us to classify the fermionic invertible symmetry defects of the (2+1)D topological phase described by a modular tensor category $\mathcal{C}$, in terms of a pair

$$(a, \rho_b) \in \mathcal{A}_0 \times \text{Aut}(\mathcal{C}), \tag{22}$$

where $\mathcal{A}_0$ is a set of Abelian bosons with $\mathbb{Z}_2$ fusion rule, and $\text{Aut}(\mathcal{C})$ is a set of automorphisms of $\mathcal{C}$.

### 2.3.1 Derivation for the canonical form of the fermionic invertible defect

Let us now derive the canonical form of the fermionic invertible defect described above. First, suppose that a given fermionic invertible defect $U$ induces a permutation action of anyons denoted as $\rho$. The permutation action $\rho$ does not give an automorphism of modular tensor category $\mathcal{C}$ in general, since it can shift the self-statistics of the anyons.

---

[3]Note that this statement is up to stacking a (1+1)D spin invertible phase to the defect, whose partition function is given by $\text{Arf}(\xi)$ with $\xi$ the spin structure of the defect.

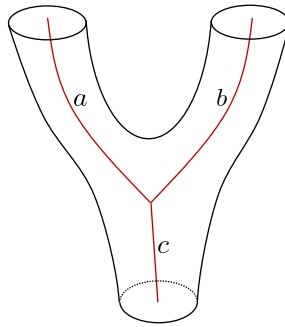

Figure 7: Bordism between disks with anyon insertions that correspond to the fusion of anyons.

While $U$ gives a gapped interface between (2+1)D topological phase $\mathcal{C}$ and itself, it is convenient to regard $U$ as a fermionic gapped boundary of the bosonic phase $\mathcal{C} \boxtimes \overline{\mathcal{C}}$, which is obtained by folding a theory $\mathcal{C}$ along the interface. On this gapped boundary of the folded theory $\mathcal{C} \boxtimes \overline{\mathcal{C}}$, the anyons in the form of $\{a, \overline{\rho(a)}\}$ for $a \in \mathcal{C}$ are condensed. In general, for a fermionic gapped boundary of the bosonic topological phase, either a boson or fermion can be condensed on the boundary. See Appendix B for a general discussion for fermionic gapped boundary of (2+1)D bosonic TQFT in terms of the Lagrangian algebra. As described in Appendix B, when the condensed anyon is a fermion (resp. boson), the Wilson line of the condensed anyon terminates on the boundary leaving the fermion parity odd (resp. even) state on the boundary.

For each anyon $a \in \mathcal{C}$, one can consider a Hilbert space of the bosonic theory $\mathcal{C} \boxtimes \overline{\mathcal{C}}$ on the disk, where an anyon $(a, \overline{\rho(a)})$ is located in the bulk of the disk, and the boundary of the disk supports a fermionic gapped boundary of $\mathcal{C} \boxtimes \overline{\mathcal{C}}$ obtained from $U$ by folding. This disk Hilbert space is one-dimensional, and let us write the state as $|D^2, a\rangle$. The fermion parity of the state is given by $\theta_a / \theta_{\rho(a)} \in \{\pm 1\}$. Then, let us consider anyons $a, b, c \in \mathcal{C}$ with non-empty fusion vertex $N_{a,b}^c > 0$. One can then take a bordism interpolating between the two disks with $(a, \overline{\rho(a)}), (b, \overline{\rho(b)})$ insertions and a single disk with $(c, \overline{\rho(c)})$ insertion, see Fig. 7. This bordism is regarded as a unitary operator between disk Hilbert spaces $|D^2, a\rangle \otimes |D^2, b\rangle$ and $|D^2, c\rangle$ which must preserve the fermion parity (i.e., preserves $\mathbb{Z}_2$ grading of spin theory). So, equating the fermion parity of the two states, we get

$$\frac{\theta_a}{\theta_{\rho(a)}} \frac{\theta_b}{\theta_{\rho(b)}} = \frac{\theta_c}{\theta_{\rho(c)}}, \quad \text{when } N_{a,b}^c > 0. \tag{23}$$

Physically, this equation tells that the fermion parity is invariant under fusion of the condensed anyons in the bulk. This implies that the phase $\theta_a / \theta_{\rho(a)} \in \{\pm 1\}$ for a generic anyon $a \in \mathcal{C}$ can be expressed as the mutual braiding between a specific Abelian anyon $v$,

$$\frac{\theta_a}{\theta_{\rho(a)}} = M_{v,a}. \tag{24}$$

This Abelian anyon $v$ satisfies the $\mathbb{Z}_2$ fusion rule $v^2 = 1$, since $v^2$ is a transparent particle in a modular tensor category $M_{v^2,a} = (M_{v,a})^2 = 1$, and must be a trivial particle. The above transformation $\rho$ between anyons satisfying Eq. (24) was studied in the context of duality of (2+1)D topological phases in [43].

Next, we show that the Abelian anyon $v$ must be a boson. To see this, we recall that the chiral central charge (framing anomaly) of the (2+1)D bosonic topological phase is given by the formula

$$e^{\frac{2\pi i}{8} c_-} = \frac{1}{\mathcal{D}} \sum_{a \in \mathcal{C}} d_a^2 \theta_a. \tag{25}$$

where the sum is over all anyons $a \in \mathcal{C}$. By the action of the fermionic defect $U$ on the modular tensor category $\mathcal{C}$, the chiral central charge is transformed to [43]

$$e^{\frac{2\pi i}{8}c_-} = \frac{1}{\mathcal{D}}\sum_{a \in \mathcal{C}}d_{\rho(a)}^2\theta_{\rho(a)} = \frac{1}{\mathcal{D}}\sum_{a \in \mathcal{C}}d_a^2\theta_a M_{v,a} = \frac{1}{\mathcal{D}}\sum_{a \in \mathcal{C}}d_{v\times a}^2\frac{\theta_{v\times a}}{\theta_v} = \frac{1}{\theta_v}\cdot\frac{1}{\mathcal{D}}\sum_{a \in \mathcal{C}}d_a^2\theta_a, \quad (26)$$

so the chiral central charge $e^{\frac{2\pi i}{8}c_-}$ is shifted by $(\theta_v)^{-1}$ by the action of $U$. Since the defect $U$ must preserve $e^{\frac{2\pi i}{8}c_-}$ to serve as the gapped interface of the bosonic topological phase $\mathcal{C}$, we have to require $\theta_v = 1$, i.e., $v$ must be a boson.

Now, one can consider the gauged Gu-Wen SPT defect $U_{\xi,v}$ out of the Abelian boson $v$, where $\xi$ is the spin structure of the fermionic defect. Then, let us consider a composite defect $V = U_{\xi,v} \times U$ obtained by fusing $U$ and the gauged Gu-Wen SPT defect. Let us write the permutation action on anyons induced by $V$ as $\rho_b$. Since the gauged Gu-Wen SPT defect shifts the spins of the anyon $a \in \mathcal{C}$ by $M_{v,a}$, one can see that the permutation action $\rho_b$ preserves the spins of the anyons, $\theta_{\rho_b(a)} = \theta_a$ for $a \in \mathcal{C}$.

In Appendix B, we show that $V$ defines a bosonic gapped interface that does not depend on spin structure of the interface. We hence get the desired expression $U = U_{\xi,v} \times V$ in terms of the bosonic invertible defect $V$ and the gauged Gu-Wen SPT defect $U_{\xi,v}$.

## 2.4 Fusion rule of fermionic invertible defects

Here we describe the fusion rule of the fermionic invertible defects of the bosonic topological phases, based on the canonical form $(a, \rho_b) \in \mathcal{A}_0 \times \text{Aut}(\mathcal{C})$ of the fermionic invertible defects discussed above.

We start with considering the fusion rule among the gauged Gu-Wen SPT defects. As we described earlier, two identical gauged Gu-Wen defects fuse into a trivial defect, $U_{\xi,a} \times U_{\xi,a} = 1$. In the following, we summarize the fusion outcome for $U_{\xi,a} \times U_{\xi,a'}$ with $a \neq a'$. The detailed computations of the fusion rules are relegated to Appendix A.

- When the mutual braiding between $a, a'$ is trivial $M_{a,a'} = 1$, the fusion rule is given by

$$U_{\xi,a} \times U_{\xi,a'} = U_{\xi,a\times a'} \times V_{a,a'}, \quad (27)$$

where $V_{a,a'}$ is a bosonic invertible defect with the $\mathbb{Z}_2$ fusion rule, defined as

$$V_{a,a'}(\Sigma) = \frac{1}{|H^1(\Sigma, \mathbb{Z}_2)|}\sum_{\Gamma,\Gamma' \in H^1(\Sigma, \mathbb{Z}_2)}W_a(\gamma)W_{a'}(\gamma')\exp\left(i\pi\int_\Sigma \Gamma \cup \Gamma'\right), \quad (28)$$

with $\gamma, \gamma' \in H_1(\Sigma, \mathbb{Z}_2)$ the Poincaré dual of $\Gamma, \Gamma'$ respectively. For example, let us consider the (2+1)D $\mathbb{Z}_2 \times \mathbb{Z}_2$ toric code and suppose that $a, a'$ are electric particles $e, e'$ for each $\mathbb{Z}_2$ gauge field respectively. Then, the defect $V_{a,a'}$ corresponds to an invertible defect given by an insertion of the topological action for the (1+1)D $\mathbb{Z}_2 \times \mathbb{Z}_2$ SPT phase along the defect [33]. In that case, the defect $V_{e,e'}$ acts on the anyons of $\mathbb{Z}_2 \times \mathbb{Z}_2$ toric code as $m \to me', m' \to m'e$, while leaving $e, e'$ invariant.

- When the mutual braiding between $a, a'$ is non-trivial $M_{a,a'} = -1$, the fusion rule is given by

$$U_{\xi,a} \times U_{\xi,a'} = U_{\xi,a'} \times V_{a\times a'} \times \text{Arf}(\xi), \quad (29)$$

where $V_{a\times a'}$ is a bosonic invertible defect with the $\mathbb{Z}_2$ fusion rule, defined as

$$V_{a\times a'}(\Sigma) = \frac{1}{\sqrt{|H^1(\Sigma, \mathbb{Z}_2)|}}\sum_{\Gamma \in H^1(\Sigma, \mathbb{Z}_2)}W_{a\times a'}(\gamma), \quad (30)$$

with $\gamma \in H_1(\Sigma, \mathbb{Z}_2)$ the Poincaré dual of $\Gamma$. For example, let us consider the (2+1)D $\mathbb{Z}_2$ toric code and suppose that $a, a'$ are electric and magnetic particle of the $\mathbb{Z}_2$ toric code respectively. Then, the $V_{a \times a'}$ is an invertible defect that induces $e \leftrightarrow m$ permutation action of anyons [33, 34].

Then, one can immediately obtain the fusion rule for the generic fermionic invertible defects expressed as the canonical form $(a, \rho_b)$. Let us write the defect that corresponds to the form $(a, \rho_b)$ as $U_{\xi,a} \times V(\rho_b)$, where $V(\rho_b)$ is a bosonic defect that induces the automorphism $\rho_b$. Then, the fusion of defects $(a, \rho_b)$, $(a', \rho_b')$ is computed as

$$U_{\xi,a} \times V(\rho_b) \times U_{\xi,a'} \times V(\rho_b') = (U_{\xi,a} \times U_{\xi,\rho_b(a')}) \times V(\rho_b \circ \rho_b'). \tag{31}$$

Since the fusion between the gauged Gu-Wen defects can be computed following Eq. (28), Eq. (29), one can obtain the fusion outcome in the canonical form as

$$U_{\xi,a} \times V(\rho_b) \times U_{\xi,a'} \times V(\rho_b') = \begin{cases} U_{\xi,a \times \rho_b(a')} \times V_{a,a'} \times V(\rho_b \circ \rho_b'), & \text{when } M_{a,\rho_b(a')} = 1, \\ U_{\xi,\rho_b(a')} \times V_{a \times a'} \times V(\rho_b \circ \rho_b') \times \text{Arf}(\xi), & \text{when } M_{a,\rho_b(a')} = -1. \end{cases} \tag{32}$$

## 2.5 Invertible symmetry defects of a (2+1)D bosonic topological phase stacked with an atomic insulator

So far, we studied the non-topological defect of the (2+1)D bosonic topological phase where the physical fermions are introduced solely at the defect. Here we describe invertible topological defects of the (2+1)D fermionic topological phase, obtained by stacking the trivial atomic insulator with the (2+1)D bosonic topological phase. Such a fermionic topological phase is described by a super-modular category $\mathcal{C} \boxtimes \{1, \psi\}$, where $\mathcal{C}$ is a modular tensor category for the bosonic phase, and the category $\{1, \psi\}$ with a physical fermion $\psi$ represents the trivial atomic insulator (i.e., trivial fermionic invertible phase).

Similar to the case of the invertible fermionic defects of the bosonic theory, one can again show that the generic invertible topological defect of the theory $\mathcal{C} \boxtimes \{1, \psi\}$ can be expressed as the fusion $U_{\xi,a} \times V(\rho_b)$, where $U_{\xi,a}$ is the gauged Gu-Wen defect with $a \in \mathcal{C}$, and the spin structure $\xi$ induced from the spin structure of the whole spacetime. $\rho_b \in \text{Aut}(\mathcal{C})$ denotes the invertible symmetry defect of the bosonic theory $\mathcal{C}$. This implies that the invertible symmetry of the theory $\mathcal{C} \boxtimes \{1, \psi\}$ also has the canonical form in terms of a pair

$$(a, \rho_b) \in \mathcal{A}_0 \times \text{Aut}(\mathcal{C}), \tag{33}$$

where $\mathcal{A}_0$ is a set of Abelian bosons with $\mathbb{Z}_2$ fusion rule.

The derivation of the canonical form can be done in parallel with the discussion in Sec. 2.3. Firstly, by folding the theory along the defect, we regard the defect as a gapped boundary of the fermionic theory $\mathcal{C} \boxtimes \overline{\mathcal{C}} \boxtimes \{1, \psi\}$, where $\{1, \psi\}$ is again understood as a trivial atomic insulator. On this gapped boundary, the anyons in the form of $a \times \overline{\rho(a)}$ is condensed for all $a \in \mathcal{C}$, where $\rho$ denotes the action of the defect on the label of the anyons, $\overline{\rho(a)} \in \overline{\mathcal{C}} \boxtimes \{1, \psi\}$.

Let us define $\sigma(a) \in \mathbb{Z}_2$ such that one can write $\rho(a) = a' \times \psi^{\sigma(a)}$ for some $a' \in \mathcal{C}$. This $\mathbb{Z}_2$ number $\sigma(a)$ denotes the fermion parity of the disk Hilbert space of the fermionic theory $\mathcal{C} \boxtimes \overline{\mathcal{C}} \boxtimes \{1, \psi\}$ with insertion of a single anyon $a \times \overline{\rho(a)}$ in its bulk, where its boundary is realized by the gapped boundary. According to the same logic as Sec. 2.3, the fermion parity of the disk Hilbert space can again be expressed as

$$(-1)^{\sigma(a)} = M_{v,a}, \tag{34}$$

where $v \in \mathcal{C}$ is an Abelian anyon with the $\mathbb{Z}_2$ fusion rule, $v^2 = 1$.

One can further show that $v$ must be a boson, by utilizing the algebraic description of the fermionic gapped boundary in terms of the Lagrangian algebra described in [44].[4] Then, one can define a gauged Gu-Wen defect $U_{\xi,v}$ that acts on the anyons $a \in \mathcal{C}$ as

$$U_{\xi,v} : a \to a \times (v \times \psi)^{\sigma(a)}. \tag{35}$$

For a given defect $V(\rho)$ of the fermionic theory $\mathcal{C} \boxtimes \{1, \psi\}$ with the action $\rho(a) = a' \times \psi^{\sigma(a)}$ (where $a' \in \mathcal{C}$), the composite defect $V(\rho_b) := U_{\xi,v} \times V(\rho)$ induces the action $\rho_b : a \to a' \times v^{\sigma(a)}$ that defines an element of $\mathrm{Aut}(\mathcal{C})$. This shows the canonical form of the invertible defect of the fermionic theory $\mathcal{C} \boxtimes \{1, \psi\}$. The fusion rule of these defects are exactly the same as Sec. 2.4.

# 3 Logical gate of Pauli stabilizer code stacked with an atomic insulator

In Sec. 2.5, we studied the invertible symmetry of the (2+1)D bosonic topological phase stacked with a trivial atomic insulator. Here, we discuss the application of these symmetry defects to the logical gates of the Pauli stabilizer models with physical fermions.

Concretely, we explicitly construct a non-Pauli Clifford logical gate for the (2+1)D $\mathbb{Z}_2$ toric code and double semion model stacked with a (2+1)D ancilla trivial atomic insulator. This logical gate corresponds to the action of the gauged Gu-Wen SPT defect discussed in Sec. 2.5, and induces the permutation of anyons that cannot be realized without the physical fermions, which leads to non-trivial action on the Pauli logical gate. For example, our logical gate in the $\mathbb{Z}_2$ toric code acts as a $CZ$ gate on the code space.

## 3.1 CZ logical gate of $\mathbb{Z}_2$ toric code

We consider a 2d square lattice on a torus $\Sigma$, with a qubit and a complex fermion on each edge. We express a single complex fermion on an edge $e$ in terms of a pair of Majorana fermions $\gamma_e, \gamma_e'$. For convenience, we put one Majorana fermion on the left side of the edge and the other on the right, as described in Fig. 8. The Hamiltonian is given by a simple stacking of (2+1)D $\mathbb{Z}_2$ toric code and a trivial atomic insulator,

$$H = -\sum_v \left( \prod_{v \subset e} X_e \right) - \sum_p \left( \prod_{p \supset e} Z_e \right) - \sum_e i\gamma_e \gamma_e'. \tag{36}$$

The ground state of the $\mathbb{Z}_2$ toric code on a torus stores two logical qubits, where the Pauli logical gates are given by

$$\overline{Z}_1 = \prod_{e \subset C_x} Z_e, \qquad \overline{X}_1 = \prod_{e \subset C_y'} X_e, \qquad \overline{Z}_2 = \prod_{e \subset C_y} Z_e, \qquad \overline{X}_2 = \prod_{e \subset C_x'} X_e. \tag{37}$$

where the product of $Z$ in $\overline{Z}_1$ is taken over edges supported on a closed loop $C_x$ of the lattice extended in $x$ direction, while the product of $X$ in $\overline{X}_2$ is over the edges cutting a closed loop $C_x'$

---

[4]This can be seen by noting that $(v, 1, m) \in \mathcal{C} \boxtimes \overline{\mathcal{C}} \boxtimes D(\mathbb{Z}_2)$ has trivial mutual braiding with all condensed anyons in the NS (anti-periodic) sector in the form of $a \times \overline{\rho(a)}$, hence is the condensed anyon in the R (periodic) sector of the Lagrangian algebra [44]. Here, $m$ is the magnetic particle of the untwisted $\mathbb{Z}_2$ gauge theory $D(\mathbb{Z}_2)$, which is physically regarded as a vortex of $\mathbb{Z}_2^f$ symmetry. This implies that the composite anyon $(v, 1, m)$ is a boson, therefore $\theta_v = 1$.



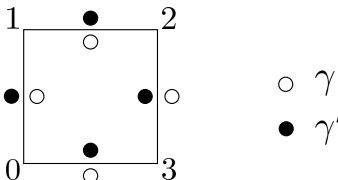

Figure 8: A pair of Majorana fermions $\gamma_e, \gamma'_e$ assigned on each edge of the 2d square lattice.

of the dual lattice. The product for $\overline{X}_1, \overline{Z}_2$ is also taken analogously. We will define a logical gate which transforms the Pauli gates as

$$\overline{X}_1 \to \overline{X}_1 \overline{Z}_2, \qquad \overline{X}_2 \to \overline{X}_2 \overline{Z}_1, \qquad \overline{Z}_1 \to \overline{Z}_1, \qquad \overline{Z}_2 \to \overline{Z}_2, \tag{38}$$

i.e., it acts as the $\overline{CZ}$ gate on the code space of the (2+1)D $\mathbb{Z}_2$ toric code. The above logical gate generates the global symmetry that induces the permutation of anyons $m \to \psi, e \to e$ of the $\mathbb{Z}_2$ gauge theory. While such a symmetry action shifting self-statistics is impossible with the bosonic unitary operators, it becomes possible when we stack the toric code with an ancilla atomic insulator, and consider the logical gate $U$ involving fermions.

The logical gate $U$ is given by

$$U = V \times \overline{Z}_1 \overline{Z}_2, \qquad V = \prod_e (\gamma_e \gamma'_e)^{\frac{1-Z_e}{2}} \times \prod_{p=(0123)} \left( \gamma_{01}^{\frac{1-Z_{01}}{2}} \gamma_{12}^{\frac{1-Z_{12}}{2}} \gamma'^{\frac{1-Z_{23}}{2}}_{23} \gamma'^{\frac{1-Z_{30}}{2}}_{30} \right), \tag{39}$$

where the vertices of a plaquette $p = (0123)$ is labeled by numbers as shown in Fig. 8. One can notice that each Majorana fermion operator $\gamma_e^{\frac{1-Z_e}{2}}$ appears twice in the expression of $V$, once in the first product over edges, and also once in the second product over plaquettes. The same also holds for $\gamma'^{\frac{1-Z_e}{2}}_e$. So, after reordering the Majorana fermions in the expression of $V$, it becomes an almost trivial operator given by $\pm 1$ sign, which does not have fermions in its expression. However, this sign depends on the eigenvalues of $Z$ on edges, so the operator $V$ can give a non-trivial logical gate acting on the code space of the toric code.

Let us study the action of $V$ on the eigenstates of $Z$ operators. For simplicity, we write the eigenvalue of each $(1 - Z_e)/2$ operator as $\alpha_e \in \{0, 1\}$, and label the eigenstate as $|\{\alpha_e\}\rangle$. One can regard $\alpha$ as a $\mathbb{Z}_2$-valued cocycle $\alpha \in Z^1(\Sigma, \mathbb{Z}_2)$ in the code space where $\delta \alpha = 0$ is satisfied on each plaquette. As we discussed earlier, the operator $V$ acts as a sign $\pm 1$ on each eigenstate,

$$V |\{\alpha_e\}\rangle = \sigma(\alpha) |\{\alpha_e\}\rangle, \quad \sigma(\alpha) \in \{\pm 1\}. \tag{40}$$

In fact, this sign $\sigma(\alpha)$ is expressed in terms of the Grassmann integral, where the Grassmann variables are assigned on edges of the square lattice. Concretely, suppose that we introduce a pair of Grassmann variables $\theta_e, \theta'_e$ on each edge, where the position of each Grassmann variable $\theta_e, \theta'_e$ are identified as that of $\gamma_e, \gamma'_e$ respectively. Then, the above sign $\sigma(\alpha)$ can be written in terms of the Grassmann integral as

$$\sigma(\alpha) = \prod_e (d\theta_e d\theta'_e)^{\alpha_e} \times \prod_{p=(0123)} \left( \theta_{01}^{\alpha_{01}} \theta_{12}^{\alpha_{12}} \theta'^{\alpha_{23}}_{23} \theta'^{\alpha_{30}}_{30} \right). \tag{41}$$

This Grassmann integral is called the Gu-Wen Grassmann integral [39, 45, 46], which is used to describe a partition function of the (1+1)D fermionic Gu-Wen SPT phase. As we will review in Appendix D, the Grassmann integral has the quadratic property that

$$\sigma(\alpha)\sigma(\alpha') = \sigma(\alpha + \alpha')(-1)^{\int_\Sigma \alpha \cup \alpha'}, \tag{42}$$

within the subspace with zero flux where $\delta\alpha = \delta\alpha' = 0$ is satisfied. Also, when $\alpha$ is given by a coboundary $\alpha = \delta\widehat{v}$ where $\widehat{v} \in C^0(\Sigma, \mathbb{Z}_2)$ is a 0-cochain which is nonzero only on a single vertex $v$, one can check that [46]

$$\sigma(\delta\widehat{v}) = 1. \tag{43}$$

This property ensures that the operator $U$ commutes with the star operator $\prod_{v \subset e} X_e$ of the Hamiltonian within the subspace with zero flux $\delta\alpha = 0$. This can be seen by

$$U\left(\prod_{v \subset e} X_e\right) U\left(\prod_{v \subset e} X_e\right) |\{\alpha_e\}\rangle = \sigma(\alpha)\sigma(\alpha + \delta\widehat{v}) |\{\alpha_e\}\rangle = \sigma(\delta\widehat{v}) |\{\alpha_e\}\rangle = |\{\alpha_e\}\rangle. \tag{44}$$

Also, $U$ obviously commutes with the plaquette operator of the toric code, as well as the fermionic terms for the Hamiltonian of the atomic insulator. So, $U$ indeed gives the logical gate.

Using the above quadratic property, one can compute the commutation relation between $U$ and Pauli logical gates. $U$ obviously commutes with $\overline{Z}_1, \overline{Z}_2$ gates, and the commutation relation between $\overline{X}$ gates within the code space are given by

$$\begin{aligned}
U\overline{X}_1 U\overline{X}_1 |\{\alpha_e\}\rangle &= -\sigma(\alpha)\sigma(\alpha + \widehat{C}'_y) |\{\alpha_e\}\rangle \\
&= (-1)^{1 + \int_\Sigma \alpha \cup \widehat{C}'_y} \times \sigma(\widehat{C}'_y) |\{\alpha_e\}\rangle \\
&= (-1)^{1 + \int_{C_y} \alpha} \times \sigma(\widehat{C}'_y) |\{\alpha_e\}\rangle \\
&= -\overline{Z}_2 \times \sigma(\widehat{C}'_y) |\{\alpha_e\}\rangle \\
&= \overline{Z}_2 |\{\alpha_e\}\rangle,
\end{aligned} \tag{45}$$

where $\widehat{C}'_y \in Z^1(\Sigma, \mathbb{Z}_2)$ is a 1-cocycle which is nonzero on the edges cutting the loop $C'_y$. In the last equation we used $\sigma(C_y) = -1$, which can be checked by explicit computation of the Grassmann integral. By similar computation, we also get

$$U\overline{X}_2 U\overline{X}_2 |\{\alpha_e\}\rangle = \overline{Z}_1 |\{\alpha_e\}\rangle. \tag{46}$$

This shows the action of $U$ as the $\overline{CZ}$ gate in Eq. (38).

## 3.2 SWAP logical gate of double semion model

Using the same method as the case of $\mathbb{Z}_2$ toric code, one can also obtain the logical gate of the double semion model that corresponds to the permutation of anyons $s \leftrightarrow \bar{s}$. To describe the double semion model, we utilize the same lattice model as presented in Sec. 2.1. That is, we have a four-dimensional qudit on each edge of the square lattice, characterized by generalized $\mathbb{Z}_4$ Pauli operators $Z, X$ satisfying $Z^4 = X^4 = 1, ZX = iXZ$. In order to implement the logical gate, we also introduce an additional complex fermion on each edge. The Hamiltonian is then given by

$$H = -\sum_v A_v - \sum_p B_p - \sum_e C_e - \sum_e i\gamma_e \gamma'_e, \tag{47}$$

where the definitions of each term is described in Fig. 5. The ground state of the $\mathbb{Z}_2$ toric code on a torus stores two logical qubits, where the Pauli logical gates are given by

$$\overline{Z}_1 = W_s(C_x), \qquad \overline{X}_1 = W_s(C_y), \qquad \overline{Z}_2 = W_{\bar{s}}(C_x), \qquad \overline{X}_2 = W_{\bar{s}}(C_y), \tag{48}$$

where $W_s, W_{\bar{s}}$ are the line operators for a semion and anti-semion respectively, which are given by the string operators for $em, em^3$ particles of $\mathbb{Z}_4$ toric code as shown in Fig. 6. The logical

gate is then given by

$$U = V \times \overline{Z}_1 \overline{Z}_2 \overline{X}_1 \overline{X}_2, \qquad V = \prod_e (\gamma_e \gamma'_e)^{\frac{1-Z_e^2}{2}} \times \prod_{p=(0123)} \left( \gamma_{01}^{\frac{1-Z_{01}^2}{2}} \gamma_{12}^{\frac{1-Z_{12}^2}{2}} \gamma_{23}'^{\frac{1-Z_{23}^2}{2}} \gamma_{30}'^{\frac{1-Z_{30}^2}{2}} \right). \quad (49)$$

One can compute the action of this logical gate on the eigenstate of $Z^2$ operators on edges. We write the eigenvalue of each $(1-Z_e^2)/2$ operator as $\alpha_e \in \{0,1\}$, and write some eigenstate with the eigenvalues $\{\alpha_e\}$ as $|\{\alpha_e\}\rangle$. One can regard $\alpha$ as flat $\mathbb{Z}_2$ gauge field $\alpha \in Z^1(\Sigma, \mathbb{Z}_2)$. The operator $V$ again acts on $|\{\alpha_e\}\rangle$ by a phase $\sigma(\alpha)$. The commutation relations between the Pauli logical gate within the code space is given by

$$\begin{aligned}
U \overline{X}_1 U \overline{X}_1 |\{\alpha_e\}\rangle &= -\sigma(\alpha)\sigma(\alpha + \widehat{C}_y') |\{\alpha_e\}\rangle \\
&= (-1)^{1+\int_\Sigma \alpha \cup \widehat{C}_y'} \times \sigma(\widehat{C}_y') |\{\alpha_e\}\rangle \\
&= (-1)^{1+\int_{C_y} \alpha} \times \sigma(\widehat{C}_y') |\{\alpha_e\}\rangle \\
&= -\overline{X}_1 \overline{X}_2 \times \sigma(\widehat{C}_y') |\{\alpha_e\}\rangle \\
&= \overline{X}_1 \overline{X}_2 |\{\alpha_e\}\rangle ,
\end{aligned} \quad (50)$$

where we note that the Wilson line for $\alpha$ along $y$ direction is given by the product of $Z^2$ operator which represents the composite particle $s\bar{s}$, so it can be expressed as $\overline{X}_1 \overline{X}_2$. Based on the similar computations, we obtain

$$U \overline{X}_2 U \overline{X}_2 = \overline{X}_1 \overline{X}_2, \qquad U \overline{Z}_1 U \overline{Z}_1 = \overline{Z}_1 \overline{Z}_2, \qquad U \overline{Z}_2 U \overline{Z}_2 = \overline{Z}_1 \overline{Z}_2, \quad (51)$$

which means that $U$ acts as the logical SWAP gate on the code space. Note that the above action on the Pauli operators corresponds to the exchange of anyons $s \leftrightarrow \bar{s}$.

# 4 Time-reversal symmetry defect with fermions: Inflow of exotic invertible phase

Here, we consider invertible fermionic interface between two (2+1)D topological phases $\mathcal{C}$ and $\mathcal{C}'$, in the case where the (2+1)D phases are defined on the boundary of a (3+1)D bosonic invertible topological phase in the bulk. We note that we do not require $\mathcal{C} = \mathcal{C}'$, while in Sec. 2 we studied the defect that interpolates a (2+1)D phase $\mathcal{C}$ and itself. When the fermionic interface is realized as a termination of the bulk interface on the boundary, one can have an anomalous action of the fermionic interface which cannot be realized in a stand-alone (2+1)D system not coupled with the bulk.

The anomalous action of the fermionic interface can be understood as follows. First, even when the interface is realized as the termination on the boundary, the relation Eq. (23) is still valid for the invertible map $\rho$ from the anyons of $\mathcal{C}$ to those of $\mathcal{C}'$ induced by the gapped interface. Hence, we can again write the shift of the self-statistics of anyons as the mutual braiding,

$$\frac{\theta_a}{\theta_{\rho(a)}} = M_{\nu,a}, \quad (52)$$

with $\nu$ an Abelian anyon with $\mathbb{Z}_2$ fusion rule, and $a \in \mathcal{C}, \rho(a) \in \mathcal{C}'$. As we have seen in Sec. 2.3, such a map $\rho$ between anyons shifts chiral central charge $\exp(2\pi i c_-/8)$ by $(\theta_\nu)^{-1}$, so the interface of a purely (2+1)D phase not coupled with the bulk must have $\theta_\nu = 1$, i.e., $\nu$ must be a boson. In that case, one can fuse the gauged Gu-Wen SPT defect $U_{\xi,\nu}$ with the

fermionic gapped interface to define a bosonic interface $V$ between $\mathcal{C}$ and $\mathcal{C}'$. Noting that $V$ is invertible, $V$ must induce an automorphism between $\mathcal{C}$ and $\mathcal{C}'$, so actually we have $\mathcal{C} = \mathcal{C}'$ up to automorphism. Hence, the invertible fermionic interface of a purely (2+1)D phase is always realized as an invertible fermionic defect of $\mathcal{C}$, which has been studied in Sec. 2.

Meanwhile, the shift of chiral central charge by the interface is possible when the (2+1)D phase is defined on the boundary of the (3+1)D Walker-Wang model. In that case, it is realized as a gapped interface that connects the Walker-Wang models with distinct surface topological order. Such a fermionic interface shifting the chiral central charge is referred to as being anomalous.

In this section, we study a curious example of the anomalous fermionic interface, where the interface is associated with an orientation-reversal of spacetime, i.e., the interface lies between $\mathcal{C}$ and its orientation-reversal $\overline{\mathcal{C}}$. We will see that such an interface can be realized by an exactly solvable Hamiltonian model in (3+1)D with spatial reflection symmetry along the plane. We take the Walker-Wang model with surface topological order $\mathcal{C} = \mathrm{U}(1)_2$, and the reflection plane works as interpolating between the Walker-Wang model with $\mathcal{C} = \mathrm{U}(1)_2$ and that with orientation-reversal $\overline{\mathcal{C}} = \mathrm{U}(1)_{-2}$. The reflection plane supports the fermionic interface shifting chiral central charge $c_-$ by $-2$. The global symmetry of the model is the spatial reflection symmetry together with $\mathbb{Z}_2^f$ fermion parity defined on the reflection plane.

Interestingly, the bulk of this system is regarded as a non-trivial (3+1)D invertible topological phase based on the above global symmetry. We will see that the model generates the $\mathbb{Z}_8$ classification of the (3+1)D invertible phase, by showing that the (2+1)D boundary for eight copies of this model becomes a trivial gapped phase by the interactions on the boundary respecting the global symmetry.

This model is described by effective invertible TQFT referred to as an exotic invertible phase in [36,37], and we discuss the relation between our model and the effective field theory later in this section.

## 4.1 Exotic invertible phase with reflection symmetry: Lattice model

Here, we construct an exactly solvable model for the fermionic interface of the Walker-Wang model with spatial reflection symmetry. The spatial reflection plane supports the fermionic interface in the model, which interpolates the $\mathrm{U}(1)_2$ topological order and its orientation reversal on the boundary. The schematic figure for the (3+1)D phase realized in our model is described in Fig. 9. Since the reflection plane supports fermions, the global symmetry of the (3+1)D phase is the reflection symmetry and the "subsystem" fermion parity $\mathbb{Z}_2^f$ localized on the reflection plane. As we will see later, the (3+1)D phase in our model realizes a non-trivial invertible topological phase protected by the combination of the global symmetries, and it generates the $\mathbb{Z}_8$ classification of the invertible topological phase.

### 4.1.1 Review: Construction of $\mathrm{U}(1)_2$ Walker-Wang model in (3+1)D

Our model utilizes a Hamiltonian model for $\mathrm{U}(1)_2$ Walker-Wang model constructed in [47], so let us review the construction here. To describe a $\mathrm{U}(1)_2$ Walker-Wang model, we start with a Walker-Wang model based on $\mathbb{Z}_4^{[1]}$ TQFT. Here, $\mathbb{Z}_4^{[1]}$ TQFT is a (2+1)D Abelian TQFT with fusion group $\mathbb{Z}_4$ generated by a semion $s$ with trivial $F$ symbol. This theory is not modular since the anyon $s^2$ is transparent, and the Walker-Wang model based on $\mathbb{Z}_4^{[1]}$ has a bulk topological order given by (3+1)D $\mathbb{Z}_2$ gauge theory. We define a Hamiltonian on a 3d cubic lattice with a four-dimensional qudit on each edge of the lattice, characterized by generalized $\mathbb{Z}_4$ Pauli operators $Z, X$ obeying the $\mathbb{Z}_4$ clock and shift algebra

$$Z^4 = X^4 = 1, \qquad ZX = iXZ. \tag{53}$$

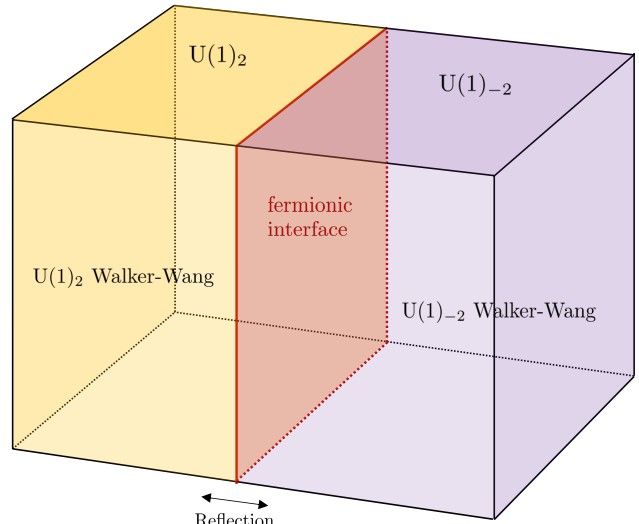

Figure 9: Schematic figure for the (3+1)D state realized in our model. The state has a spatial reflection symmetry and a subsystem $\mathbb{Z}_2^f$ symmetry that acts on fermions localized on the reflection plane.

The Hamiltonian of $\mathbb{Z}_4^{[1]}$ Walker-Wang model takes the form of

$$\widetilde{H} = -\sum_v A_v - \sum_p \widetilde{B}_p + \text{h.c.}, \tag{54}$$

where $A_v$ is defined on each vertex of a cubic lattice, and $\widetilde{B}_p$ is on each plaquette. The definition of these terms are given in Fig. 10. As shown in [47], the Hamiltonian $\widetilde{H}$ has a (3+1)D topological order given by $\mathbb{Z}_2$ gauge theory. The electric particle is generated by string operators $C_e$ defined on edges as Fig. 11. These string operators satisfy

$$\widetilde{B}_p^2 = \prod_{e \in p} C_e, \qquad C_e^2 = 1. \tag{55}$$

Also, the string operators satisfy the commutation relation between the Hamiltonian

$$C_e A_v = \begin{cases} -A_v C_e, & \text{if } v \in e, \\ +A_v C_e, & \text{if } v \notin e, \end{cases} \qquad [C_e, \widetilde{B}_p] = [C_e, C_{e'}] = 0, \tag{56}$$

which means that the operator $C_e$ on $e = \langle vv' \rangle$ creates a pair of electric particles on vertices $v$ and $v'$, characterized by $A_v = A_{v'} = -1$.

Starting with the $\mathbb{Z}_4^{[1]}$ Walker-Wang Hamiltonian $\widetilde{H}$, one can obtain a lattice model for the (3+1)D invertible phase by condensing the electric particle. The condensed Hamiltonian has the form of

$$H_{\text{U}(1)_2} = \sum_p \widetilde{B}_p + \text{h.c.} - \sum_e C_e \tag{57}$$

which is obtained by adding the hopping terms $C_e$ of the electric particle to $\widetilde{H}$, and picking the terms of $\widetilde{H}$ that commute with the hopping terms. The Hamiltonian $H_{\text{U}(1)_2}$ does not host topological order in the bulk, but it turns out that the (2+1)D boundary of the (3+1)D bulk hosts a chiral topological order realized by $\text{U}(1)_2$, where a semion becomes a deconfined anyon excitation on the boundary.

Let us now describe the bulk-boundary Hamiltonian for $H_{\text{U}(1)_2}$. For simplicity, we consider a "half-infinite" geometry where the model is defined on the region $z \le 0$ in the 3d Euclidean

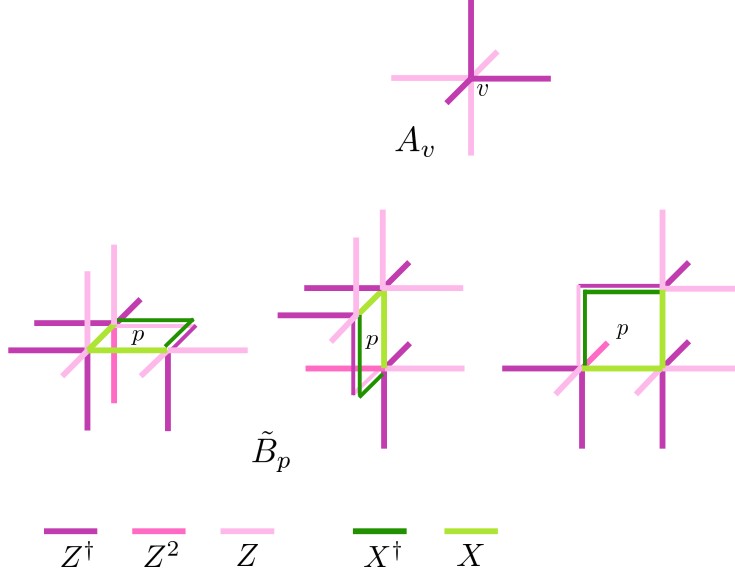

Figure 10: The Hamiltonian for $\mathbb{Z}_4^{[1]}$ Walker-Wang model.

space, and the boundary is supported on the 2d plane $z = 0$. The bulk-boundary Hamiltonian is then given by the same form as the case without boundary,

$$H_{\mathrm{U}(1)_2} = \sum_p \widetilde{B}_p + \mathrm{h.c.} - \sum_e C_e \tag{58}$$

where we simply define the boundary Hamiltonian as the truncation of the terms in the bulk. To see the topological order on the (2+1)D boundary at $z = 0$, we explicitly describe the operators that creates a pair of semions on the boundary. Let us consider an operator $\mathcal{X}_e$ for each edge $e = \langle vv' \rangle$, which works on the boundary as a hopping term for a semion that creates a pair of anyons at the adjacent vertices $v$, $v'$. This operator $\mathcal{X}_e$ is defined in both bulk and boundary, and depends on the direction of the edge $e = \langle vv' \rangle$, satisfying $\mathcal{X}_{\langle vv' \rangle} = \mathcal{X}_{\langle v'v \rangle}^\dagger = \mathcal{X}_e$. $\mathcal{X}_e$ is given by Fig. 11 in the bulk, and $\mathcal{X}_e$ on the boundary is defined by truncation. $\mathcal{X}_e$ satisfies the following properties,

$$Z_e \mathcal{X}_e = i \mathcal{X}_e Z_e, \qquad Z_e \mathcal{X}_{e'} = \mathcal{X}_{e'} Z_e, \tag{59}$$

$$\widetilde{B}_p = \mathcal{X}_{12} \mathcal{X}_{23} \mathcal{X}_{34} \mathcal{X}_{41} Z_O^2, \tag{60}$$

$$\mathcal{X}_e^2 = C_e, \tag{61}$$

where $O$ labels an edge determined relatively by the configuration of the plaquette $p$ as described in Fig. 11. Note that due to the truncation on the boundary, the operators $\mathcal{X}_e$ on the boundary $z = 0$ satisfies

$$\widetilde{B}_p = \mathcal{X}_{12} \mathcal{X}_{23} \mathcal{X}_{34} \mathcal{X}_{41}, \tag{62}$$

which means that a closed line operator for $\mathcal{X}_e$ on the boundary commutes with the Hamiltonian, while it does not in the bulk. This line operator for $\mathcal{X}_e$ is identified as an Wilson line of the semion $s$ on the boundary. Due to the relation $\mathcal{X}_e^2 = C_e$, fusing two semions gives a trivial anyon condensed on both bulk and boundary.

### 4.1.2 Hamiltonian for the exotic invertible phase

Now we obtain a reflection symmetric model starting with a $\mathrm{U}(1)_2$ Walker-Wang Hamiltonian $H_{\mathrm{U}(1)_2}$. We prepare our model on the cubic lattice of a 3d Euclidean space with a reflection

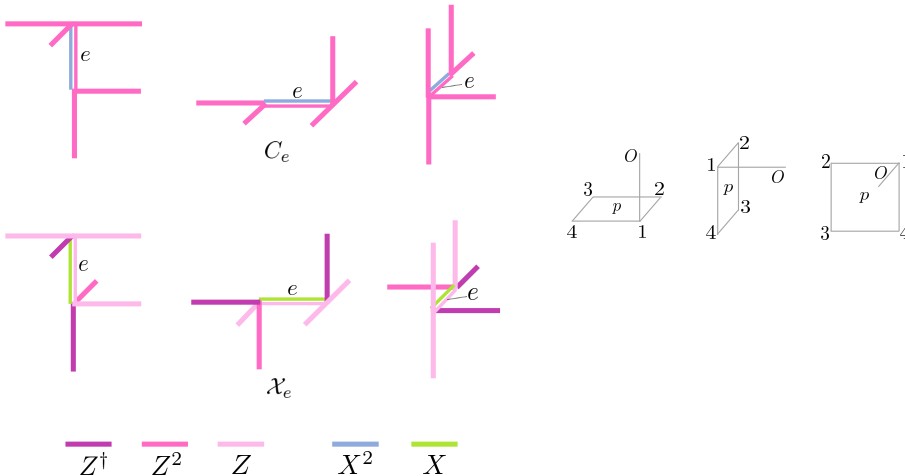

Figure 11: The operators that correspond to short string operators for anyons.

plane at $x = 0$. Since the reflection plane supports a fermionic defect, we introduce a complex fermion on each vertex $v$ on the reflection plane $x = 0$. A complex fermion on a vertex $v$ is represented by a pair of Majorana fermions $\gamma_v, \gamma'_v$. Also, at the reflection plane $x = 0$, we introduce a pair of $\mathbb{Z}_4$ qudits $\{X_{e;l}, Z_{e;l}\}, \{X_{e;r}, Z_{e;r}\}$ for each edge $e$. The reflection symmetry acts trivially on operators as

$$\mathsf{R}(\gamma_v) = \gamma_v, \qquad \mathsf{R}(\gamma'_v) = \gamma'_v, \qquad \mathsf{R}(X_e) = X_{\mathsf{R}(e)}, \qquad \mathsf{R}(Z_e) = Z_{\mathsf{R}(e)}, \tag{63}$$

where we define $\mathsf{R}(X_{e;l}) := X_{e;r}$, $\mathsf{R}(Z_{e;l}) := Z_{e;r}$.

The reflection symmetric Hamiltonian can be obtained by preparing the Hamiltonian $H_{\mathrm{U}(1)_2;l}$ on the left region $x \leq 0$ with boundary at $x = 0$ and its reflection partner $H_{\mathrm{U}(1)_{-2};r}$ on the right $x \geq 0$, and then gluing them along the reflection plane $x = 0$. The Hamiltonian $H_{\mathrm{U}(1)_2;l}$ is simply given by the truncation of $H_{\mathrm{U}(1)_2}$ for the boundary Hamiltonian near $x = 0$, and the boundary qudits of $H_{\mathrm{U}(1)_2;l}$ are given by $\{X_{e;l}, Z_{e;l}\}$. We then define $H_{\mathrm{U}(1)_{-2};r} := \mathsf{R}(H_{\mathrm{U}(1)_2;l})$ for the Hamiltonian at $x \geq 0$.

We glue the Hamiltonians at $x = 0$ by condensing the pair of semions from each boundary on the left and the right, combined with a local fermion on the reflection plane. This condensation is done by adding a term on the reflection plane $x = 0$ in the form of

$$\sum_{\substack{e=\langle vv' \rangle \\ \in \{x=0\}}} i\gamma_v \gamma_{v'} \mathcal{X}_{e;l} \mathcal{X}_{e;r}, \tag{64}$$

which is a hopping term of a pair of semions together with a Majorana fermion on the reflection plane. One can check that the operators in the form of $i\gamma_v \gamma_{v'} \mathcal{X}_{e;l} \mathcal{X}_{e;r}$ commutes with each other, since $\mathcal{X}_e$ satisfies the commutation relation

$$\mathcal{X}_e \mathcal{X}_{e'} = \begin{cases} \pm i \mathcal{X}_{e'} \mathcal{X}_e, & \text{if } e, e' \text{ share a single vertex,} \\ \mathcal{X}_{e'} \mathcal{X}_e, & \text{if } e, e' \text{ don't share a vertex, or } e = e', \end{cases} \tag{65}$$

where the sign of $\pm i$ depends on the detail of the position of $e, e'$ but not important for the discussion here.

We note that simply adding the term $i\gamma_v \gamma_{v'} \mathcal{X}_{e;l} \mathcal{X}_{e;r}$ to the reflection plane does not give a commuting Hamiltonian, since $i\gamma_v \gamma_{v'} \mathcal{X}_{e;l} \mathcal{X}_{e;r}$ is not commuting with the $\widetilde{B}_p, C_e$ terms on the boundary of $H_{\mathrm{U}(1)_2;l}, H_{\mathrm{U}(1)_{-2};r}$. So, we have to modify the Hamiltonians $H_{\mathrm{U}(1)_2;l}, H_{\mathrm{U}(1)_{-2};r}$ near

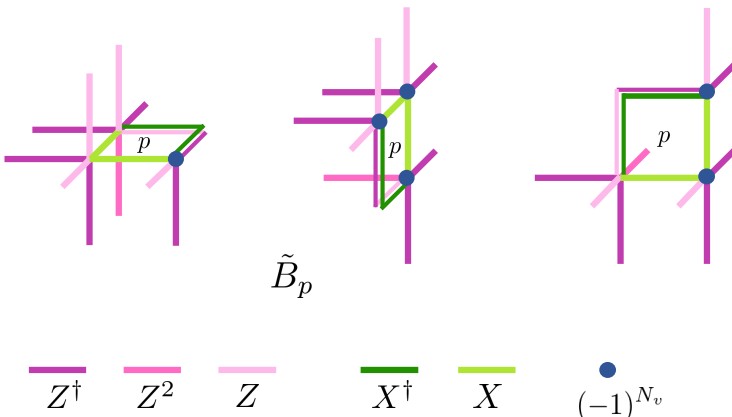

$$\widetilde{B}_p$$

$$Z^\dagger \qquad Z^2 \qquad Z \qquad\qquad X^\dagger \qquad X \qquad (-1)^{N_v}$$

Figure 12: The plaquette terms $\widetilde{B}_p$ on the left region touching the reflection plane. The blue dots denote the fermion parity operator $(-1)^{N_v} = i\gamma_v\gamma'_v$ supported on the reflection plane. The Hamiltonians on the right are defined as the reflection partners of them.

the boundary so that they commute with $i\gamma_v\gamma_{v'}\mathcal{X}_{e;l}\mathcal{X}_{e;r}$. The $C_e$ operators of $H_{U(1)_2;l}, H_{U(1)_{-2};r}$ are modified to $C'_e$ for $e = \langle vv' \rangle$ given by

$$C'_e = C_e P_v P_{v'}, \tag{66}$$

where $P_v$ is defined as

$$\begin{cases} P_v = 1 & \text{if } v \text{ is not on the reflection plane } x = 0 \\ P_v = (-1)^{N_v} & \text{if } v \text{ is on the reflection plane } x = 0 \end{cases} \tag{67}$$

Also, the $\widetilde{B}_p$ are also modified near the reflection plane as shown in Fig. 12. Let us denote the modified Hamiltonians as $H'_{U(1)_2;l}, H'_{U(1)_{-2};r}$. One can see that these Hamiltonians are now commutative with each term in the form of $i\gamma_v\gamma_{v'}\mathcal{X}_{e;l}\mathcal{X}_{e;r}$. Then, the Hamiltonian for the exotic invertible topological phase on the 3d Euclidean space is expressed as

$$H = H'_{U(1)_2;l} + H'_{U(1)_{-2};r} - \left( \sum_{\substack{e = \langle vv' \rangle \\ \in \{x=0\}}} i\gamma_v\gamma_{v'}\mathcal{X}_{e;l}\mathcal{X}_{e;r} + \text{h.c.} \right). \tag{68}$$

One can see that the reflection plane does not carry ground state degeneracy if the reflection plane is supported on a torus, so the reflection plane does not host topological order. This implies that the bulk of this (3+1)D model is invertible. The detailed discussion is relegated to Appendix C.

### 4.1.3 Surface topological order of exotic invertible phase

Let us now consider the (2+1)D boundary of the exotic invertible phase by constructing a bulk-boundary system. As reviewed in Sec. 4.1.1, we put the (3+1)D bulk on a region $z \le 0$, and the boundary is supported on a 2d plane $z = 0$. The bulk-boundary Hamiltonian is then simply defined by truncating the terms near the boundary,

$$H = H'_{U(1)_2;l} + H'_{U(1)_{-2};r} - \left( \sum_{\substack{e = \langle vv' \rangle \\ \in \{x=0\}}} i\gamma_v\gamma_{v'}\mathcal{X}_{e;l}\mathcal{X}_{e;r} + \text{h.c.} \right). \tag{69}$$

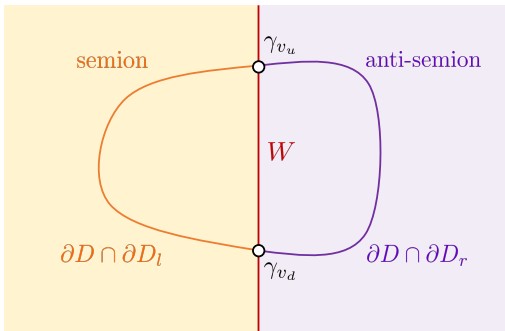

Figure 13: The Wilson line operator of the surface $U(1)_2$ topological order crossing the 1d domain domain wall. When the line operator crosses through the domain wall, the intersection supports a Majorana fermion (white dot).

The (2+1)D boundary of this system realizes a fermionic gapped domain wall separating $U(1)_2$ and $U(1)_{-2}$ topological order as schematically shown in Fig. 9. The domain wall transforms the semion of $U(1)_2$ to an anti-semion of $U(1)_{-2}$, which is regarded as the time-reversal action $T : U(1)_2 \to U(1)_{-2}$. To find this action of the domain wall on the anyon, we explicitly describe a line operator of the semion crossing through the domain wall on the reflection plane.

For this purpose, we recall that the plaquette operator $\widetilde{B}_p$ on the boundary is regarded as a small loop of the Wilson line for the semion, as shown in Eq. (62). So, one can obtain the form of the line operator surrounding a closed region by multiplying the $\widetilde{B}_p$ operators inside the region. Then, let us pick a disk region $D$ on the 2d boundary, where $D$ is separated by the reflection plane into two regions $D_l$, $D_r$. Let us denote the interval for the 1d domain wall $W$ contained in the region $D$. If we take the product of $\widetilde{B}_p$ operators inside $D$, we then have

$$\prod_{p \subset D} \widetilde{B}_p = \prod_{\substack{e \subset \partial D_l \\ \text{anti-clockwise}}} \mathcal{X}_e \prod_{\substack{e \subset \partial D_r \\ \text{clockwise}}} \mathcal{X}_e = \left( \prod_{\substack{e \subset \partial D \cap \partial D_l \\ \text{anti-clockwise}}} \mathcal{X}_e \prod_{\substack{e \subset \partial D \cap \partial D_r \\ \text{clockwise}}} \mathcal{X}_e \right) \cdot \prod_{e \in W} \mathcal{X}_{e;l} \mathcal{X}_{e;r} . \tag{70}$$

Here, the term inside the parenthesis is supported on $\partial D$ and regarded as a line operator of an anyon, while the last term is the unwanted contribution on the domain wall. See Fig. 13 for the configuration of operators. Note that one can eliminate the terms on the domain walls by multiplying the terms of the Hamiltonian with the form $i\gamma_v \gamma_{v'} \mathcal{X}_{e;l} \mathcal{X}_{e;r}$ on the domain wall. Then we obtain the closed line operator on $\partial D$ as

$$\prod_{p \subset D} \widetilde{B}_p \cdot \prod_{e \in W} i\gamma_v \gamma_{v'} \mathcal{X}_{e;l} \mathcal{X}_{e;r} \propto \left( \prod_{\substack{e \subset \partial D \cap \partial D_l \\ \text{anti-clockwise}}} \mathcal{X}_e \prod_{\substack{e \subset \partial D \cap \partial D_r \\ \text{clockwise}}} \mathcal{X}_e \right) \cdot \gamma_{v_d} \gamma_{v_u} , \tag{71}$$

where $v_d, v_u$ are two vertices located at the intersection between $\partial D$ and the domain wall. This means that a pair of semions from $U(1)_2$, $U(1)_{-2}$ can terminate at the domain wall, leaving a Majorana fermion $\gamma$ at the termination. In other words, the fermionic domain wall transforms the anyons according to the action $T : U(1)_2 \to U(1)_{-2}$, shifting the spin of the anyon by $1/2$ on the fermionic domain wall.

## 4.2 Eight copies of exotic invertible phases: $\mathbb{Z}_8$ classification

Here, we argue that our model generates the $\mathbb{Z}_8$ classification of the (3+1)D invertible phase with spatial reflection and $\mathbb{Z}_2^f$ fermion parity on the reflection plane. We demonstrate the $\mathbb{Z}_8$

classification by explicitly taking eight copies of our models, and describing a way of anyon condensation on (2+1)D boundary respecting the symmetry that turns the boundary theory into a trivial gapped state. The argument is in a similar spirit to the seminal work by Fidkowski and Kitaev [48], which showed that the boundary of eight copies of Majorana chains is turned into a trivial gapped state by quartic interactions of fermions on boundary preserving time-reversal symmetry.

On the left of the reflection plane, (2+1)D boundary of eight copies of exotic invertible phases is given by $(U(1)_2)^8$ topological order, which has eight semions $s_j$ for $j = 1, \ldots, 8$. The topological order can be transformed into a trivial gapped phase by condensing the following anyons of the Lagrangian subgroup $G_{FK}$ generated by [5]

$$
\begin{aligned}
\{1, & s_1 s_2 s_3 s_4, s_5 s_6 s_7 s_8, s_1 s_2 s_5 s_6, \\
& s_3 s_4 s_7 s_8, s_2 s_3 s_6 s_7, s_1 s_4 s_5 s_8, s_1 s_3 s_5 s_7, \\
& s_3 s_4 s_5 s_6, s_1 s_2 s_7 s_8, s_2 s_3 s_5 s_8, s_1 s_4 s_6 s_7, \\
& s_2 s_4 s_6 s_8, s_1 s_3 s_6 s_8, s_2 s_4 s_5 s_7, s_1 s_2 s_3 s_4 s_5 s_6 s_7 s_8 \}\,.
\end{aligned}
\tag{72}
$$

These anyons are obviously bosons since a quartet of semions has topological twist $i^4 = 1$. This group is $(\mathbb{Z}_2)^4$, generated by four anyons

$$
s_1 s_2 s_3 s_4, s_5 s_6 s_7 s_8, s_1 s_2 s_5 s_6, s_1 s_3 s_5 s_7\,.
\tag{73}
$$

Since these four bosons mutually braid trivially with each other, these anyons generate Lagrangian subgroup by fusion. Though one can suppress the topological order by condensing anyons (72) away from the reflection plane, it is still non-trivial whether one can trivially gap out the whole system in the presence of a fermionic gapped interface on the reflection plane.

To see what happens near the reflection plane, let us consider the eight copies of (2+1)D boundary folded at the reflection plane, see Fig. 14. The folded theory is given by $(U(1)_2)^8 \times (U(1)_2)^8$, where each $(U(1)_2)^8$ represents the left or right of the reflection plane. Let us write the semions of each $(U(1)_2)^8$ as $\{s_j\}$ and $\{s'_j\}$ for $1 \le j \le 8$. The reflection plane is then regarded as a fermionic gapped boundary of $(U(1)_2)^8 \times (U(1)_2)^8$, given by condensing fermions $\{s_j s'_j\}$ for each $j$ on the boundary. In our lattice model, the line operator of a fermion $s_j s'_j$ terminates at the boundary with a Majorana fermion $\gamma_j$. Let us denote this fermionic boundary condition of $(U(1)_2)^8 \times (U(1)_2)^8$ as $\mathcal{B}_f$.

Meanwhile, we can condense the bosons listed in Eq. (72) away from the reflection plane for each $(U(1)_2)^8$. One can perform this anyon condensation away from the reflection plane, making a bosonic gapped boundary of $(U(1)_2)^8 \times (U(1)_2)^8$ characterized by Lagrangian subgroup $G_{FK} \times G_{FK}$. This process obviously respects the spatial reflection symmetry. Let us denote this bosonic boundary condition as $\mathcal{B}_{FK}$. Then, the folded system is given by a thin slab of $(U(1)_2)^8 \times (U(1)_2)^8$ sandwiched by the boundary conditions $\mathcal{B}_{FK}$ and $\mathcal{B}_f$, see Fig. 14.

It turns out that this thin slab carries the nontrivial ground state degeneracy, and we need to further turn on interaction terms on the slab to obtain a trivial gapped state. To count the ground state degeneracy of the slab, suppose that the slab is prepared on a space $S^1 \times I$ with $I$ an interval.[6] The slab $S^1 \times I$ is regarded as a sphere $S^2$ with two punctures. By shrinking the punctures into points, the spatial geometry reduces to $S^2$ with insertions of two Lagrangian algebra anyons $\overline{\mathcal{L}}_{FK} = \mathcal{L}_{FK}$ and $\mathcal{L}_f$ which correspond to each boundary condition on two boundary circles respectively [44, 49]. Hence, the dimension of the Hilbert space on the slab is given

---

[5]Here the subscript refers to Fidkowski-Kitaev, since the quartet of semions represented here has the exactly same form of the interaction terms of eight Majorana fermions introduced in [48].

[6]Since we introduce fermions on one boundary of the slab, we need to introduce spin structure on the boundary $S^1$. For simplicity, let us assume that $S^1$ on one boundary of the slab is equipped with NS spin structure.

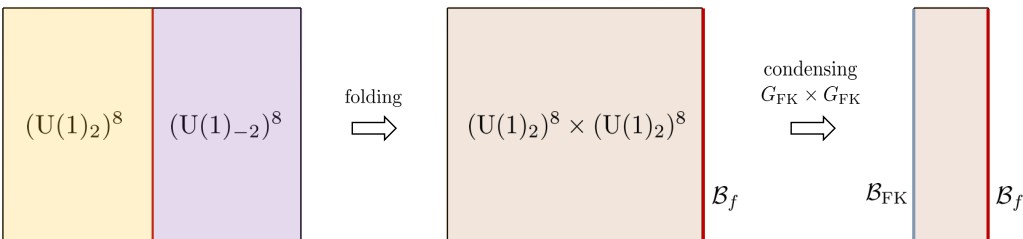

Figure 14: For eight copies of boundaries of exotic invertible phases, we fold the system at the reflection plane and consider $(U(1)_2)^8 \times (U(1)_2)^8$ topological order. By performing anyon condensation of $G_{FK} \times G_{FK}$ away from the reflection plane, one obtains a thin slab of $(U(1)_2)^8 \times (U(1)_2)^8$ bounded by two distinct bosonic and fermionic gapped boundary conditions.

by the number of anyons in the group $\mathcal{L}_{FK} \cap \mathcal{L}_f$. Writing $\psi_j := s_j s_j'$, the group $\mathcal{L}_{FK} \cap \mathcal{L}_f$ is generated by the anyons

$$\psi_1 \psi_2 \psi_3 \psi_4, \psi_5 \psi_6 \psi_7 \psi_8, \psi_1 \psi_2 \psi_5 \psi_6, \psi_1 \psi_3 \psi_5 \psi_7, \tag{74}$$

and has order $2^4 = 16$. The slab hence carries the 16-fold ground state degeneracy, and each ground state is characterized by the eigenvalue of small line operators of anyons $\mathcal{L}_{FK} \cap \mathcal{L}_f$ extended along the interval $I$ terminating on the boundaries. The degeneracy can be lifted by further condensing the sixteen particles of $\mathcal{L}_{FK} \cap \mathcal{L}_f$ for both bulk and boundary of the slab. When the width of the slab is taken thin enough, the anyon condensation of the above particles is performed locally, and this process respects the reflection symmetry since the above anyons in Eq. (74) is reflection invariant. Physically, the condensation of anyons (74) amounts to turning on quartic interaction term of Majorana fermions on the fermionic interface which is the same as the one introduced in [48], since $\psi$ terminates at the interface leaving the Majorana fermion operator $\gamma$. It shows that the boundary for eight copies of the (3+1)D phase can be brought to a trivial gapped phase by anyon condensation on the boundary respecting the global symmetries.

## 4.3 Effective field theory of exotic invertible phase

Here we review the effective field theory for the exotic invertible phase developed in [36, 37], and then make contact with our lattice Hamiltonian model. Roughly speaking, the effective field theory for the exotic invertible phase is given by the (3+1)D Crane-Yetter-Walker-Wang TQFT [35, 50] with $U(1)_2$ topological order on its boundary, in the presence of spacetime orientation-reversing (i.e., time-reversal) defects.

The Crane-Yetter-Walker-Wang model with $U(1)_2$ on its boundary is described by a $\mathbb{Z}_2$ gauge theory with the 2-form dynamical $\mathbb{Z}_2$ gauge field $b \in Z^2(M^4, \mathbb{Z}_2)$, with the action [51, 52]

$$\exp\left(\frac{2\pi i}{4} \int_{M^4} \mathcal{P}(b)\right), \tag{75}$$

where $\mathcal{P}(b) = b \cup b - b \cup_1 \delta b$ is the Pontryagin square, which gives an element of $H^4(M^4, \mathbb{Z}_4)$.

Let us consider the time-reversal defect of this theory. Since the time-reversal conjugates the action, the theory is obviously non-invariant under the time-reversal. This can be understood as the time-reversal anomaly of the action given by a (4+1)D theory

$$\exp\left(\pi i \int_{5D} w_1 \cup b \cup b\right), \tag{76}$$

where $w_1$ is the first Stiefel-Whitney class of the spacetime manifold that represents the Poincaré dual of time-reversal defect. Though this anomaly obstructs from making $b$ dynamical while respecting time-reversal symmetry, one can cancel out this anomaly by introducing an additional fermionic theory on the time-reversal defect. To see this, let us rewrite the above (4+1)D action for the anomaly as

$$\exp\left(\pi i \int_W b \cup b\right),\tag{77}$$

where $W$ is an oriented 4-manifold given by the Poincaré dual of $w_1$. Due to the Wu formula [53], $w_2(TW) \cup b + b \cup b$ is exact as an element of $H^4(W, \mathbb{Z}_2)$. So, when the time-reversal defect $W$ is equipped with spin structure $\xi$ satisfying $\delta\xi = w_2(TW)$, one can trivialize the above anomaly by coupling the theory with spin structure of $W$. Indeed, it turns out that one can explicitly construct a local action for (2+1)D fermionic theory living on the time-reversal defect $W^3$ of $M^4$, which has exactly the same anomaly as Eq. (77) [39]. We write the action for the fermionic theory as

$$\exp\left(\pi i \int_{W^3} q_\xi(b)\right),\tag{78}$$

whose partition function is given by $\pm 1$ for any closed manifold $W^3$. After all, the combined action

$$\exp\left(\frac{2\pi i}{4} \int_{M^4} \mathcal{P}(b) + \pi i \int_{W^3} q_\xi(b)\right),\tag{79}$$

preserves the time-reversal symmetry. In [37], it is shown that this $\mathbb{Z}_2$ gauge theory defines a (3+1)D invertible phase with time-reversal symmetry and spin structure ($\mathbb{Z}_2^f$ symmetry) on the time-reversal defect.

Note that our lattice Hamiltonian model simulates the (3+1)D topological action described above. In our model, the orientation-reversing defect is realized as a spatial reflection plane that interpolates the domain of $U(1)_2$ Walker-Wang model and its reflection partner. Then, the fermionic wave function supported on it is regarded as a theory $\exp\left(\pi i \int_{W^3} q_\xi(b)\right)$ in the action, which is based on "subsystem" $\mathbb{Z}_2^f$ symmetry that corresponds to the spin structure $\xi$ of the defect.

While we have shown that our lattice model generates the $\mathbb{Z}_8$ classification of the invertible phase with spatial reflection and subsystem $\mathbb{Z}_2^f$ symmetry, one can also obtain the $\mathbb{Z}_8$ classification of the invertible phase at the level of effective field theory. This can be seen by evaluating the above path integral Eq. (79) on a closed 4-manifold equipped with the spin structure on the orientation-reversing defect, and checking that it is always given by $\exp(2\pi i \nu/8)$ with $\nu \in \mathbb{Z}_8$, see [37] for the detailed discussions. For example, it is known that the partition function of the above theory evaluated on an oriented manifold is given by $\exp(2\pi i \sigma(M^4)/8)$ with $\sigma(M^4)$ the signature of the manifold, so the partition function on e.g., $\mathbb{CP}^2$ is given by $\exp(2\pi i/8)$.

While our theory requires the spin structure $\xi$ of the orientation-reversing defect $W^3$, $\xi$ can also be regarded as a certain geometric structure of the whole spacetime manifold. The spin structure $\xi$ on the orientation-reversing defect satisfies $\delta\xi = w_2(TW^3)$, and $w_2(TW^3)$ can be represented by some 1-cycle of the orientation-reversing defect $W^3$. It was shown in [37] that when we regard this 1-cycle as an element of $Z_1(M^4, \mathbb{Z}_2)$ by embedding in the whole spacetime $M^4$, this 1-cycle represents the class $(w_1 w_2)(TM^4)$. Hence, the choice of $\xi$ in tern specifies the trivialization of the third cohomology class $\delta\xi' = w_1 w_2(TM^4)$. This spacetime structure characterized by $\delta\xi' = w_1 w_2(TM^4)$ is called Wu structure, which is regarded as a 2-group which is the mixture between $\mathbb{Z}_2$ 1-form symmetry and the spacetime Lorentz symmetry, where the third Postnikov class is given by $w_1 w_2(TM^4)$ [36]. The above effective field theory

Eq. (79) is regarded as the invertible TQFT that depends on Wu structure of a spacetime manifold.

## 4.4 Discussions: Relation to non-trivial QCA and topological superconductor

Here, let us make a comment on the unitary operator that disentangles our 3d lattice model. When a gapped state with spatial reflection symmetry is short-range entangled, one can think of applying finite-depth local unitary (FDLU) in a reflection symmetric fashion to disentangle the bulk of the state, which obviously respects the reflection symmetry of the state. After performing the FDLU, the original state reduces to the state localized in the vicinity of the reflection plane, since the state away from the reflection plane is decoupled. This process is indeed useful for classifying the SPT phase with spatial reflection symmetry, since it reduces the original state with spatial symmetry to the lower-dimensional state localized at the reflection plane, where the reflection symmetry now acts internally. This idea is called dimensional reduction [54], and used to classify a large class of SPT phases with generic crystalline symmetry. For example, the (3+1)D bosonic invertible phase with reflection symmetry is classified by $\mathbb{Z}_2 \times \mathbb{Z}_2$ [55,56], and one of generators for $\mathbb{Z}_2$ reduces to the (2+1)D SPT phase with internal $\mathbb{Z}_2$ symmetry by dimension reduction, which is also classified by $H^3(B\mathbb{Z}_2, U(1)) = \mathbb{Z}_2$.

Meanwhile, the bulk of our 3d lattice model cannot be disentangled by the FDLU,[7] hence one cannot carry out the dimension reduction to reduce the state to the lower-dimensional (2+1)D fermionic phase. This statement is based on a conjecture that the unitary that disentangles the Walker-Wang model with chiral surface topological order gives a non-trivial quantum cellular automata (QCA) [57]. Here, non-trivial QCA means a locality-preserving unitary that cannot be prepared by FDLU. An explicit construction of QCA that disentangles the $U(1)_2$ Walker-Wang model utilized in our model is given in [47], so we instead have to apply this QCA for disentangling the bulk of the exotic invertible phase.

Let us also comment on the relation between our model and the (3+1)D fermionic invertible phase with reflection symmetry. While our 3d model has fermions localized at the 2d reflection plane, one can introduce additional complex fermions away from the reflection plane, and make them simply define a trivial atomic insulator. Then, we can obtain a model for the (3+1)D fermionic invertible phase with reflection symmetry (acting as $R^2 = 1$). The (3+1)D fermionic invertible phase with reflection is classified by $\mathbb{Z}_{16}$ [40,54]. The resulting phase after introducing additional fermions to our model gives the $\nu = 2$ phase in the $\mathbb{Z}_{16}$ classification. It should be noted that this $\nu = 2$ phase is also realized by an isolated location of (2+1)D $\mathbb{Z}_2 \times \mathbb{Z}_2^f$ SPT phase classified by $\mathbb{Z}_8$ [58] at the reflection plane, with a trivial atomic insulator elsewhere [54]. The reflection symmetry acts as internal $\mathbb{Z}_2$ symmetry at the reflection plane, and this picture of $\nu = 2$ phase seems to imply that one can perform dimension reduction of our exotic invertible phase in the presence of additional fermions. It is likely that the bulk of the exotic invertible phase can be turned to a trivial bosonic product state with trivial atomic insulator by fermionic FDLU.

So, after introducing additional fermions to the bulk, we expect that the bosonic QCA that disentangles the $U(1)_2$ Walker-Wang model is regarded as fermionic FDLU, i.e., trivial as a "fermionic QCA" which should be defined as the locality-preserving unitary of fermionic Fock space respecting $\mathbb{Z}_2^f$ symmetry. Meanwhile, the fermionic surface topological order $U(1)_2 \times U(1)_{-1}$ does not admit a fermionic gapped boundary, though it does not carry chiral central charge.[8] It seems incompatible with the above statement that $U(1)_2$ Walker-Wang

---

[7]Here, by FDLU we mean the unitary circuit which is exactly local, where we do not allow the circuit to have exponentially decaying tails.

[8]It can be checked as follows. When the fermionic theory admits a gapped boundary, one of its bosonic modular extensions must have a bosonic gapped interface with untwisted $\mathbb{Z}_2$ gauge theory $D(\mathbb{Z}_2)$ [44]. Meanwhile, the bosonic modular extension of $U(1)_2 \times U(1)_{-1}$ with $c_- = 0 \mod 8$ is given by $U(1)_2 \times U(1)_{-4}$, which does not admit

model with local fermions can be disentagled by the fermionic FDLU, since such a FDLU would make it possible to define a commuting projector Hamiltonian for the $U(1)_2 \times U(1)_{-1}$ topological order, starting with the $U(1)_2$ Walker-Wang model and disentangling the bulk by fermionic FDLU. It contradicts with the widely held belief that the topological order that does not admit gapped boundary cannot have a realization by a commuting projector Hamiltonian model. It would be interesting to resolve this problem by developing the understanding for fermionic analogue of QCA.

## Acknowledgements

The author thanks Yu-An Chen and Sahand Seifnashri for useful conversations. The author thanks Po-Shen Hsin and Nat Tantivasadakarn for comments on a draft.

**Funding information** The author is supported by the JQI postdoctoral fellowship at the University of Maryland.

## A  Detailed analysis of the gauged Gu-Wen SPT defects

In this appendix, we perform explicit computations to derive the algebraic properties of gauged Gu-Wen SPT defect

$$U_{\xi,a}(\Sigma) = \frac{1}{\sqrt{|H^1(\Sigma,\mathbb{Z}_2)|}} \text{Arf}(\xi) \sum_{\Gamma \in H^1(\Sigma,\mathbb{Z}_2)} W_a(\gamma) \exp\left(i\pi \int_\Sigma q_\xi(\Gamma)\right), \tag{A.1}$$

with an Abelian boson $a$ with $\mathbb{Z}_2$ fusion rule, where $\gamma \in H_1(\Sigma,\mathbb{Z}_2)$ is the Poincaré dual of $\Gamma$.

### A.1  Fusion rules

Here we compute the fusion rules described in the main text. First, let us derive the $\mathbb{Z}_2$ fusion rule of $U_{\xi,a}$. It can be computed as

$$\begin{aligned}
U_{\xi,a}(\Sigma) \times U_{\xi,a}(\Sigma) &= \frac{1}{|H^1(\Sigma,\mathbb{Z}_2)|} \sum_{\Gamma,\Gamma' \in H^1(\Sigma,\mathbb{Z}_2)} W_a(\gamma+\gamma') \exp\left(i\pi \int_\Sigma q_\xi(\Gamma+\Gamma') + \Gamma \cup \Gamma'\right) \\
&= \frac{1}{|H^1(\Sigma,\mathbb{Z}_2)|} \sum_{\Gamma,\Gamma' \in H^1(\Sigma,\mathbb{Z}_2)} W_a(\gamma') \exp\left(i\pi \int_\Sigma q_\xi(\Gamma') + \Gamma \cup \Gamma'\right) \\
&= \sum_{\Gamma' \in H^1(\Sigma,\mathbb{Z}_2)} W_a(\gamma') \exp\left(i\pi \int_\Sigma q_\xi(\Gamma')\right) \delta(\Gamma') \\
&= 1.
\end{aligned} \tag{A.2}$$

Next, let us compute the fusion rule $U_{\xi,a} \times U_{\xi,a'}$ when $a \neq a'$. As we have seen in the main text, the fusion outcome depends on the mutual braiding $M_{a,a'} = \pm 1$. We show the following fusion rule,

- When the mutual braiding between $a, a'$ is trivial $M_{a,a'} = 1$, the fusion rule is given by

$$U_{\xi,a} \times U_{\xi,a'} = U_{\xi,a\times a'} \times V_{a,a'}, \tag{A.3}$$

___
a gapped interface with $D(\mathbb{Z}_2)$.

where $V_{a,a'}$ is a bosonic invertible defect with the $\mathbb{Z}_2$ fusion rule, defined as

$$V_{a,a'}(\Sigma) = \frac{1}{|H^1(\Sigma,\mathbb{Z}_2)|} \sum_{\Gamma,\Gamma' \in H^1(\Sigma,\mathbb{Z}_2)} W_a(\gamma) W_{a'}(\gamma') \exp\left( i\pi \int_\Sigma \Gamma \cup \Gamma' \right), \qquad (A.4)$$

with $\gamma, \gamma' \in H_1(\Sigma,\mathbb{Z}_2)$ the Poincaré dual of $\Gamma, \Gamma'$ respectively.

- When the mutual braiding between $a, a'$ is non-trivial $M_{a,a'} = -1$, the fusion rule is given by

$$U_{\xi,a} \times U_{\xi,a'} = U_{\xi,a'} \times V_{a \times a'} \times \mathrm{Arf}(\xi), \qquad (A.5)$$

where $V_{a \times a'}$ is a bosonic invertible defect with the $\mathbb{Z}_2$ fusion rule, defined as

$$V_{a \times a'}(\Sigma) = \frac{1}{\sqrt{|H^1(\Sigma,\mathbb{Z}_2)|}} \sum_{\Gamma \in H^1(\Sigma,\mathbb{Z}_2)} W_{a \times a'}(\gamma), \qquad (A.6)$$

with $\gamma \in H_1(\Sigma,\mathbb{Z}_2)$ the Poincaré dual of $\Gamma$.

To show Eq. (A.3), we compute the right hand side as

$$
\begin{aligned}
U_{\xi,a \times a'}(\Sigma) \times V_{a,a'}(\Sigma) =& \frac{\mathrm{Arf}(\xi)}{|H^1(\Sigma,\mathbb{Z}_2)|^{\frac{3}{2}}} \sum_\Gamma W_a(\gamma) W_{a'}(\gamma) \exp\left( i\pi \int_\Sigma q_\xi(\Gamma) \right) \\
& \times \sum_{\Gamma',\Gamma''} W_a(\gamma') W_{a'}(\gamma'') \exp\left( i\pi \int_\Sigma q_\xi(\Gamma') + q_\xi(\Gamma'') + q_\xi(\Gamma' + \Gamma'') \right) \\
=& \frac{\mathrm{Arf}(\xi)}{|H^1(\Sigma,\mathbb{Z}_2)|^{\frac{3}{2}}} \sum_{\Gamma,\Gamma',\Gamma''} W_a(\gamma + \gamma') W_{a'}(\gamma + \gamma'') \\
& \times \exp\left( i\pi \int_\Sigma q_\xi(\Gamma + \Gamma') + q_\xi(\Gamma'') + q_\xi(\Gamma' + \Gamma'') + \Gamma \cup \Gamma' \right) \\
=& \frac{\mathrm{Arf}(\xi)}{|H^1(\Sigma,\mathbb{Z}_2)|^{\frac{3}{2}}} \sum_{\Gamma,\Gamma',\Gamma''} W_a(\gamma + \gamma') W_{a'}(\gamma + \gamma'') \\
& \times \exp\left( i\pi \int_\Sigma q_\xi(\Gamma + \Gamma') + q_\xi(\Gamma + \Gamma'') + q_\xi(\Gamma + \Gamma' + \Gamma'') \right) \\
=& \frac{\mathrm{Arf}(\xi)}{|H^1(\Sigma,\mathbb{Z}_2)|^{\frac{3}{2}}} \sum_{\Gamma,\Gamma',\Gamma''} W_a(\gamma) W_{a'}(\gamma') \exp\left( i\pi \int_\Sigma q_\xi(\Gamma) + q_\xi(\Gamma') + q_\xi(\Gamma'') \right) \\
=& U_{\xi,a}(\Sigma) \times U_{\xi,a'}(\Sigma). \qquad (A.7)
\end{aligned}
$$

To show Eq. (A.5), we compute $U_{\xi,a'} \times U_{\xi,a} \times U_{\xi,a'}$ as

$$
\begin{aligned}
&(U_{\xi,a'} \times U_{\xi,a} \times U_{\xi,a'})(\Sigma) = \\
&= \frac{\text{Arf}(\xi)}{|H^1(\Sigma,\mathbb{Z}_2)|^{\frac{3}{2}}} \sum_{\Gamma,\Gamma',\Gamma''} W_a(\gamma) W_{a'}(\gamma') W_a(\gamma'') \exp\left(i\pi \int_\Sigma q_\xi(\Gamma) + q_\xi(\Gamma') + q_\xi(\Gamma'')\right) \\
&= \frac{\text{Arf}(\xi)}{|H^1(\Sigma,\mathbb{Z}_2)|^{\frac{3}{2}}} \sum_{\Gamma,\Gamma',\Gamma''} W_a(\gamma+\gamma'') W_{a'}(\gamma') \exp\left(i\pi \int_\Sigma q_\xi(\Gamma) + q_\xi(\Gamma') + q_\xi(\Gamma'') + \Gamma' \cup \Gamma''\right) \\
&= \frac{\text{Arf}(\xi)}{|H^1(\Sigma,\mathbb{Z}_2)|^{\frac{3}{2}}} \sum_{\Gamma,\Gamma',\Gamma''} W_a(\gamma+\gamma'') W_{a'}(\gamma') \exp\left(i\pi \int_\Sigma q_\xi(\Gamma+\Gamma'+\Gamma'') + \Gamma \cup (\Gamma'+\Gamma'')\right) \\
&= \frac{\text{Arf}(\xi)}{|H^1(\Sigma,\mathbb{Z}_2)|^{\frac{3}{2}}} \sum_{\Gamma,\Gamma',\Gamma''} W_a(\gamma) W_{a'}(\gamma') \exp\left(i\pi \int_\Sigma q_\xi(\Gamma+\Gamma') + \Gamma \cup \Gamma' + (\Gamma+\Gamma') \cup \Gamma''\right) \\
&= \frac{\text{Arf}(\xi)}{|H^1(\Sigma,\mathbb{Z}_2)|^{\frac{1}{2}}} \sum_{\Gamma,\Gamma'} W_a(\gamma) W_{a'}(\gamma') \exp\left(i\pi \int_\Sigma q_\xi(\Gamma+\Gamma') + \Gamma \cup \Gamma'\right) \delta(\Gamma+\Gamma') \\
&= V_{a \times a'}(\Sigma) \times \text{Arf}(\xi).
\end{aligned}
\tag{A.8}
$$

## A.2 Permutation action on anyons

Here we derive the permutation action of the gauged Gu-Wen defect $U_{\xi,a}$ on anyons $a \in \mathcal{C}$ given by

$$
\begin{cases}
p \to p, & \text{if } M_{p,a} = 1, \\
p \to p \times a, & \text{if } M_{p,a} = -1.
\end{cases}
\tag{A.9}
$$

The permutation action can be evaluated by computing the fusion outcome of $U_{\xi,a}(\Sigma) \times W_p(\gamma) \times U_{\xi,a}(\Sigma)$, where $W_p(\gamma)$ is the Wilson line operator for an anyon $p$ on the curve $\gamma$ embedded in $\Sigma$.

When $M_{p,a} = 1$, the operators $W_p$ and $U_{\xi,a}$ simply commute with each other, so we have $U_{\xi,a}(\Sigma) \times W_p(\gamma) \times U_{\xi,a}(\Sigma) = W_p(\gamma)$ and $p$ is invariant under the symmetry action. When $M_{p,a} = -1$, we have

$$
\begin{aligned}
&U_{\xi,a}(\Sigma) \times W_p(\gamma) \times U_{\xi,a}(\Sigma) = \\
&= \frac{1}{|H^1(\Sigma,\mathbb{Z}_2)|} \sum_{\Gamma',\Gamma''} W_a(\gamma') W_p(\gamma) W_a(\gamma'') \exp\left(i\pi \int_\Sigma q_\xi(\Gamma') + q_\xi(\Gamma'')\right) \\
&= \frac{1}{|H^1(\Sigma,\mathbb{Z}_2)|} \sum_{\Gamma',\Gamma''} W_p(\gamma) W_a(\gamma'+\gamma'') \exp\left(i\pi \int_\Sigma q_\xi(\Gamma'+\Gamma'') + \Gamma \cup \Gamma' + \Gamma' \cup \Gamma''\right) \\
&= \frac{1}{|H^1(\Sigma,\mathbb{Z}_2)|} \sum_{\Gamma',\Gamma''} W_p(\gamma) W_a(\gamma') \exp\left(i\pi \int_\Sigma q_\xi(\Gamma') + \Gamma \cup (\Gamma'+\Gamma'') + \Gamma' \cup \Gamma''\right) \\
&= \sum_{\Gamma'} W_p(\gamma) W_a(\gamma') \exp\left(i\pi \int_\Sigma q_\xi(\Gamma') + \Gamma \cup \Gamma'\right) \delta(\Gamma+\Gamma') \\
&= W_{p \times a}(\gamma) \times \exp\left(i\pi \int_\Sigma q_\xi(\Gamma)\right),
\end{aligned}
\tag{A.10}
$$

which shows that the anyon $p$ is permuted to $p \times a$, and the line operator is dressed with the (0+1)D spin invertible phase with the action $q_\xi(\Gamma)$ along the curve $\gamma$.

### A.3  Summing over spin structures

Here we perform the explicit computation of gauging $\mathbb{Z}_2^f$ fermion parity symmetry of the gauged Gu-Wen defect $U_{\xi,a}(\Sigma)$ to obtain a bosonic topological defect $\widetilde{U}_a(\Sigma)$,

$$\widetilde{U}_a(\Sigma) := \frac{1}{|H^1(\Sigma, \mathbb{Z}_2)|} \sum_\xi U_{\xi,a}(\Sigma). \tag{A.11}$$

The sum over the spin structures can be done by

$$
\begin{aligned}
\widetilde{U}_a(\Sigma) &= \frac{1}{|H^1(\Sigma, \mathbb{Z}_2)|^{\frac{3}{2}}} \sum_{\Gamma',\xi} \exp\left( i\pi \int_\Sigma q_\xi(\Gamma') \right) \sum_\Gamma W_a(\gamma) \exp\left( i\pi \int_\Sigma q_\xi(\Gamma) \right) \\
&= \frac{1}{|H^1(\Sigma, \mathbb{Z}_2)|^{\frac{3}{2}}} \sum_{\Gamma,\Gamma',\xi} W_a(\gamma) \exp\left( i\pi \int_\Sigma q_\xi(\Gamma + \Gamma') + \Gamma \cup \Gamma' \right) \\
&= \frac{1}{\sqrt{|H^1(\Sigma, \mathbb{Z}_2)|}} \sum_{\Gamma,\Gamma'} W_a(\gamma) \exp\left( i\pi \int_\Sigma \Gamma \cup \Gamma' \right) \delta(\Gamma + \Gamma') \\
&= \frac{1}{\sqrt{|H^1(\Sigma, \mathbb{Z}_2)|}} \sum_{\Gamma \in H^1(\Sigma, \mathbb{Z}_2)} W_a(\gamma).
\end{aligned}
\tag{A.12}
$$

So, the topological defect $\widetilde{U}_a(\Sigma)$ is the condensation defect obtained by gauging the symmetry generated by the Wilson line operator $W_a(\gamma)$ on the defect $\Sigma$ [34].

# B  Gapped domain wall and Lagrangian algebra

## B.1  Review: Lagrangian algebra anyons

We begin with a brief review for some properties of gapped boundary of a bosonic topological phase without global symmetry. See [49] for detailed descriptions.

Gapped boundary of a (2+1)D bosonic topological quantum field theory (TQFT) is algebraically described by an object called a Lagrangian algebra anyon [49, 59–61]. The idea is that when a TQFT admits a topological gapped boundary condition, we consider cutting out a solid cylinder from a spacetime 3-manifold. We introduce a gapped boundary condition on the boundary of the resulting manifold, getting a cylinder of gapped boundary. We shrink the radius of the cylinder of a gapped boundary, then it eventually becomes a topological line operator of a TQFT, see Fig. 15. So, the tube of the gapped boundary after shrinking is expressed as a sum of simple anyons in a modular tensor category $\mathcal{C}$,

$$\mathcal{L} = \bigoplus_{a \in \mathcal{C}} Z_{0a} a, \tag{B.1}$$

with non-negative integers $Z_{0a}$. This object $\mathcal{L}$ is called a Lagrangian algebra anyon. Since one can cap off the tube on the top of it and introduce gapped boundary on the cap, the tube of a gapped boundary can end at a point. It means that $\mathrm{Hom}(\mathcal{L}, 1)$ is not empty, and hence $Z_{00} > 0$. See Fig. 15. In general, the Lagrangian algebra anyon with $Z_{00} > 1$ is known to decompose into the sum of those with $Z_{00} = 1$. The simple gapped boundary condition is hence described by the Lagrangian algebra anyon satisfying $Z_{00} = 1$.

When $Z_{0a} > 0$ for some anyon $a \in \mathcal{C}$, $\mathrm{Hom}(\mathcal{L} \times a, \mathcal{L})$ is not empty since $Z_{00} > 0$. This implies that the Wilson line of $a$ can end on the tube of gapped boundary, meaning that $a$ is condensed on the boundary. So, anyons with $Z_{0a} > 0$ is physically regarded as a set of condensed anyons.

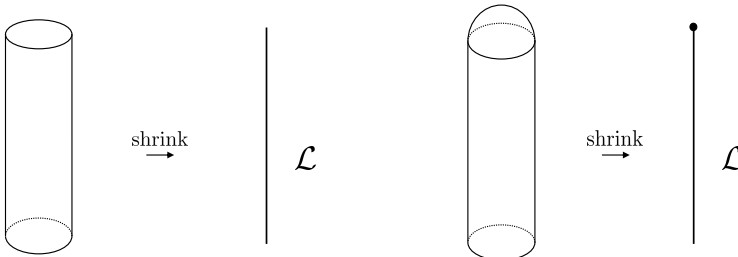

Figure 15: One can prepare a cylinder of a gapped boundary, then shrinking it results in a line operator. One can cap off the cylinder by a gapped boundary, so the line operator can end at a point, which means that $\mathrm{Hom}(\mathcal{L}, 1)$ is not empty.

$\mathcal{L}$ satisfies nice properties under modular $S, T$ transformations. The vector $\{Z_{0a}\}$ turns out to be an eigenvector of modular $S$ and $T$ matrices:

$$\sum_{b \in \mathcal{C}} S_{ab} Z_{0b} = Z_{0a}, \qquad \sum_{b \in \mathcal{C}} T_{ab} Z_{0b} = Z_{0a}. \tag{B.2}$$

Since $S$ and $T$ of modular tensor category $(ST)^3 = e^{\frac{2\pi i}{8} c_-} S^2$, the existence of the Lagrangian algebra anyon with (B.2) implies $c_- = 0 \mod 8$.

We can consider a fusion space of the Lagrangian algebra anyon $V_{\mathcal{L}}^{\mathcal{L}\mathcal{L}}$ by taking a junction of three tubes of gapped boundary. We can then talk about the $F$- and $R$-move of tubes with junctions, which turn out to be trivial:

$$(F_{\mathcal{L}}^{\mathcal{L}\mathcal{L}\mathcal{L}})_{\mathcal{L},\mathcal{L}} \cdot |\mu\rangle \otimes |\mu\rangle = |\mu\rangle \otimes |\mu\rangle, \tag{B.3}$$

$$R_{\mathcal{L}}^{\mathcal{L}\mathcal{L}} |\mu\rangle = |\mu\rangle. \tag{B.4}$$

### B.2 Fermionic gapped interface of the bosonic topological phases

Here let us provide algebraic description for the fermionic gapped interface $U$ of the bosonic topological phase represented by a modular tensor category $\mathcal{C}$. Along with this, we complete the derivation in Sec. 2.3 that the fermionic gapped interface $U$ can in general be expressed as fusion of the gauged Gu-Wen SPT defect $U_{\xi,a}$ and a bosonic invertible gapped interface $V$.

We restrict ourselves to the case that the interface $U$ is invertible, where the interface induces an invertible map between anyons $\rho$. By folding the picture along the interface, the interface reduces to the gapped boundary of the folded theory $\mathcal{C} \boxtimes \overline{\mathcal{C}}$, with the fermions introduced on the gapped boundary. In the folded picture, the anyon in the form of $(a, \overline{\rho(a)})$ in $\mathcal{C} \boxtimes \overline{\mathcal{C}}$ is condensed on the boundary.

We study the algebraic description for the fermionic gapped boundary of the theory $\mathcal{C} \boxtimes \overline{\mathcal{C}}$ in terms of the Lagrangian algebra. To construct the Lagrangian algebra anyon for a given fermionic gapped boundary, we again carve out the solid cylinder from a spacetime 3-manifold. The boundary of the carved 3-manifold then becomes a cylinder, where we introduce the fermionic gapped boundary condition. Since we introduce fermions on the boundary, the cylinder on the boundary is equipped with spin structure. By shrinking the cylinder, it becomes a topological line operator of a TQFT. The operator obtained by shrinking the cylinder is again expressed as a sum of simple anyons in $\mathcal{C} \boxtimes \overline{\mathcal{C}}$, but note that the shrunk object generally depends on the spin structure along the meridian of the cylinder, see Fig. 16. For each spin structure, it is expressed as

$$\mathcal{L}_{\mathrm{NS}} = \bigoplus_{x \in \mathcal{C} \boxtimes \overline{\mathcal{C}}} Z_x^{\mathrm{NS}} x, \qquad \mathcal{L}_{\mathrm{R}} = \bigoplus_{x \in \mathcal{C} \boxtimes \overline{\mathcal{C}}} Z_x^{\mathrm{R}} x. \tag{B.5}$$

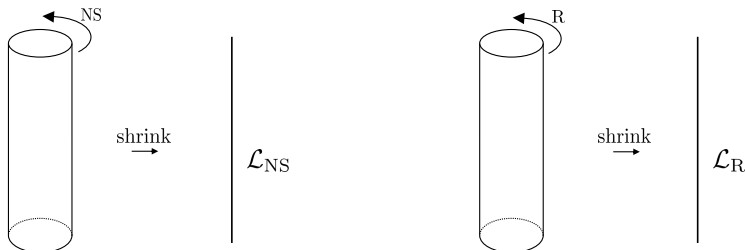

Figure 16: One can shrink the tube of gapped boundary into a line, and the resulting object depends on spin structure along the meridian.

We call the non-simple anyon $\mathcal{L}_{\text{NS}}, \mathcal{L}_{\text{R}}$ listed above the fermionic Lagrangian algebra anyon.

For the cylinder with the NS spin structure along the meridian, one can "cap off" the cylinder by a disk as shown in Fig. 15, since the spin structure extends to the disk on the top. This means that $\text{Hom}(\mathcal{L}_{\text{NS}}, 1)$ is not empty, i.e., $Z_{00}^{\text{NS}} > 0$. Also, the condensed anyons on the boundary have the form of $(a, \overline{\rho(a)})$, the line operators for anyons can terminate at the top of the capped cylinder iff they are expressed as $(a, \overline{\rho(a)})$. This means $\text{Hom}(\mathcal{L}_{\text{NS}}, x)$ is not empty iff $x = (a, \overline{\rho(a)})$ for some $a \in \mathcal{C}$, i.e., $Z_{0x}^{\text{NS}} > 0$ iff $x = (a, \overline{\rho(a)})$.

To study the properties of the Lagrangian algebra anyons, it is convenient to consider a boundary state $|\mathcal{L}_{\mu,\lambda}\rangle$ on a space $T^2$ given by considering a spacetime $T^2 \times [0, 1]$, and then equipping the spin structure $(\mu, \lambda)$ for $\mu, \lambda \in \{\text{NS}, \text{R}\}$ on one side of the boundary $T^2 \times \{1\}$. After introducing the fermionic gapped boundary on $T^2 \times \{1\}$, one defines a state $|\mathcal{L}_{\mu,\lambda}\rangle$ on $T^2 \times \{0\}$. Equivalently, one can also regard the geometry as a solid torus $D^2 \times S^1$, with a thin solid torus cut out at the center of $D^2$. So, the boundary state is thought of as an insertion of a line operator in $D^2 \times S^1$ along $S^1$ obtained by a thin tube of gapped boundary. The boundary state with the spin structure $\{\mu, \lambda\} = \{\text{NS}, \text{NS}\}$ is given by

$$|\mathcal{L}_{\text{NS,NS}}\rangle = \sum_{a \in \mathcal{C}} Z_{(a, \overline{\rho(a)})}^{\text{NS}} |(a, \overline{\rho(a)})\rangle . \tag{B.6}$$

The boundary state has an important property under the modular $S, T$ transformations. That is, once we assume that the boundary is gapped and topological, the diffeomorphism acting on the gapped boundary at $T_{\mu,\lambda}^2 \times \{1\}$ must not change the boundary state $|\mathcal{L}_{\mu,\lambda}\rangle$. Since the torus is equipped with spin structure, the diffeomorphism here means the mapping class group of $T_{\mu,\lambda}^2$ leaving spin structure invariant. For the case of $\mu = \text{NS}$, the Dehn twist $T$ along the meridian exchanges the spin structure as $T_{\text{NS,NS}}^2 \leftrightarrow T_{\text{NS,R}}^2$, so we have $T^2 |\mathcal{L}_{\text{NS,NS}}\rangle = |\mathcal{L}_{\text{NS,NS}}\rangle$. This implies that an anyon with $Z_{0x}^{\text{NS}} > 0$ must be either a boson or a fermion, since Dehn twist acts on Wilson lines as $|x\rangle \rightarrow \theta_x |x\rangle$. We then have $\theta_a / \theta_{\rho(a)} \in \{\pm 1\}$ for each $a \in \mathcal{C}$.

As we discussed in Sec. 2.3, for a given fermionic interface, one can explicitly construct a gauged Gu-Wen SPT defect $U_{\xi,\nu}$ with the Abelian boson $\nu$ that satisfies $M_{\nu,a} = \theta_a / \theta_{\rho(a)}$, where the action of the composite interface $V := U_{\xi,\nu} \times U$ induces a map $\rho_b$ that leaves the spins of anyons invariant, $\theta_{\rho_b(a)} = \theta_a$ for all $a \in \mathcal{C}$.

From now, let us study the properties of the redefined interface $V$, and show that $V$ is a bosonic interface that does not depend on the spin structure of the interface. Again, let us consider a gapped boundary of the folded theory $\mathcal{C} \boxtimes \overline{\mathcal{C}}$ by folding the geometry along the interface $V$. The anyons in the form of $(a, \overline{\rho_b(a)})$ is condensed on this gapped boundary. Let us denote the Lagrangian algebra anyon for this gapped boundary as $\mathcal{L}'_{\text{NS}}, \mathcal{L}'_{\text{R}}$. The boundary state for this boundary condition $|\mathcal{L}'_{\text{NS,NS}}\rangle$ is given in the form of

$$|\mathcal{L}'_{\text{NS,NS}}\rangle = \sum_{a \in \mathcal{C}} Z'^{\text{NS}}_{(a, \overline{\rho_b(a)})} |(a, \overline{\rho_b(a)})\rangle . \tag{B.7}$$

Since $\rho_b$ leaves the spins of anyons invariant, the anyons $(a, \overline{\rho_b(a)})$ are bosons. This implies that the above boundary state is invariant under the modular $T$ transformation along the meridian,

$$|\mathcal{L}'_{\text{NS,R}}\rangle = T |\mathcal{L}'_{\text{NS,NS}}\rangle = |\mathcal{L}'_{\text{NS,NS}}\rangle \,. \tag{B.8}$$

By studying the $S$ modular transformation of the boundary states $|\mathcal{L}'_{\text{NS,NS}}\rangle$, $|\mathcal{L}'_{\text{NS,R}}\rangle$, one can see that [44]

$$\sum_{y \in \mathcal{C} \boxtimes \overline{\mathcal{C}}} S_{xy} Z'^{\text{NS}}_y = Z'^{\text{NS}}_x \,, \qquad \sum_{y \in \mathcal{C} \boxtimes \overline{\mathcal{C}}} S_{xy} Z'^{\text{NS}}_y = Z'^{\text{R}}_x \,, \quad \text{for } x \in \mathcal{C} \boxtimes \overline{\mathcal{C}}. \tag{B.9}$$

Then, we obviously have $Z'^{\text{NS}}_x = Z'^{\text{R}}_x$ for all $x \in \mathcal{C} \boxtimes \overline{\mathcal{C}}$. This means that the fermionic Lagrangian algebra anyon for the interface $V$ is independent of the choice of the spin structure, so let us simply write $\mathcal{L}' := \mathcal{L}'_{\text{NS}} = \mathcal{L}'_{\text{R}}$, $Z'_x := Z'^{\text{NS}}_x = Z'^{\text{R}}_x$. The insensitivity of the fermionic Lagrangian algebra anyon to the spin structure implies that the gapped boundary is in fact bosonic rather than fermionic. Actually, by using that the gapped boundary is topological, one can see that $\mathcal{L}'$ satisfies all the properties of the Lagrangian algebra anyon Eq. (B.2), (B.3), (B.4) for a bosonic gapped boundary reviewed in Appendix B.2. For example, (B.3) can be derived by regarding the fusion vertex of the Lagrangian algebra anyons as a trivalent junction of tubes of gapped boundary, then noting that $F$-move can be realized by continuous deformation of the manifold that supports gapped boundary [49]. This shows that $V$ defines a bosonic gapped interface, and all the invertible fermionic gapped interface $U$ is expressed as the fusion $\text{U}(1)_{\xi,\nu} \times V$.

## C  Detailed description of the exotic invertible phase

In this appendix, we describe the detailed calculations about the lattice model of the exotic invertible phase constructed in Sec. 4.

### C.1  Reflection plane does not carry topological order

The lattice model for the exotic invertible phase has a reflection plane with a fermion. Here we argue that the reflection plane does not carry topological order, and hence defines the invertible domain wall between the $\text{U}(1)_2$ and $\text{U}(1)_{-2}$ Walker-Wang models. To see this, we consider the geometry of a 3d space where the reflection plane is located on the $yz$ plane at $x = 0$ in the 3d Euclidean space, and we take the periodic boundary condition in the $y$, $z$ directions so that the reflection plane is defined on a torus $T^2$. We take the $x$ direction as an interval $-M \le x \le M$, i.e., there are $M$ lattice spacings in the $x$ direction on both left and right side of the reflection plane.

We count the ground state degeneracy of the lattice model of the exotic topological phase in this geometry, and show that the degeneracy is carried entirely by the topological order on the (2+1)D boundary at $x = -M, M$ instead of the reflection plane. The Hamiltonian is given by Eq. (68), with a technical modification on the boundary

$$H = H'_{\text{U}(1)_2;l} + H'_{\text{U}(1)_{-2};r} - \left( \sum_{\substack{e=\langle vv' \rangle \\ \in \{x=0\}}} i\gamma_v \gamma_{v'} \mathcal{X}_{e;l} \mathcal{X}_{e;r} + \text{h.c.} \right) - \sum_{v \in \{x=-M\}} A_v^2 - \sum_{v \in \{x=M\}} A_v^2, \tag{C.1}$$

where the last two terms are additionally introduced in order to make the boundary support $\text{U}(1)_{\pm 2}$ topological order.

Suppose that there are $N$ vertices on the $yz$ plane. The degrees of freedom are the $N$ complex fermions on the reflection plane, and $(3M + 2)N$ qudits on each (left or right) side of the reflection plane. The dimension of the total Hilbert space is given by $\dim \mathcal{H} = 2^N \cdot 4^{(3M+2)N} \cdot 4^{(3M+2)N}$. To count the ground state degeneracy, we list the constraints of the stabilizers in the Hamiltonian. They are given as follows.

- There are $2(3M + 2)N$ $C'_e$ terms in the Hamiltonian $H'_{U(1)_2;l} + H'_{U(1)_{-2};r}$. Each of them gives an order two constraint $C'_e = 1$.

- There are $2(3M + 1)N$ $\widetilde{B}_p$ terms in the Hamiltonian $H'_{U(1)_2;l} + H'_{U(1)_{-2};r}$. Due to $\widetilde{B}_p^2 = \prod_{e \in p} C'_e$, each $\widetilde{B}_p = 1$ gives an order two constraint after restricting the Hilbert space to the subspace satisfying $C'_e = 1$.

- There are $2N$ $i\gamma_v \gamma_{v'} \mathcal{X}_{e;l} \mathcal{X}_{e;r}$ terms in the Hamiltonian. Due to $(i\gamma_v \gamma_{v'} \mathcal{X}_{e;l} \mathcal{X}_{e;r})^2 = C'_{e;l} C'_{e;r}$, each of them gives an order two constraint after restricting the Hilbert space to the subspace satisfying $C'_e = 1$.

- There are $2N$ $A_v^2$ terms in the last two terms of the Hamiltonian. Each of them gives an order two constraint.

Note that some of the constraints listed above are redundant, since taking the product of certain stabilizers gives the identity operator. The redundancy comes from the following $(N+2)$ relations among the stabilizers,

- For each plaquette $p$ on the reflection plane $x = 0$, we have

$$\prod_{e \in p} (i\gamma_v \gamma_{v'} \mathcal{X}_{e;l} \mathcal{X}_{e;r}) \cdot \widetilde{B}_p^l \widetilde{B}_p^r = 1 \,. \tag{C.2}$$

Since there are $N$ plaquettes on the reflection plane, there are $N$ such equations which are independent with each other.

- At the left (2+1)D boundary on $x = -M$, multiplying the terms $A_v^2, \widetilde{B}_p$ is given by

$$\prod_{v \in \{x=-M\}} A_v^2 \prod_{p \in \{x=-M\}} \widetilde{B}_p = 1 \,. \tag{C.3}$$

The similar equation also holds for the right boundary $x = M$. They give the two equations where the product of the stabilizers becomes the identity.

So, the ground state degeneracy of the Hamiltonian is given by

$$\text{GSD} = \dim \mathcal{H} \cdot \frac{1}{2^{2(3M+2)N} \cdot 2^{2(3M+1)N} \cdot 2^{2N} \cdot 2^{2N}} \cdot 2^{N+2} = 4 \,. \tag{C.4}$$

This four-fold degeneracy originates from the $U(1)_2$ topological order for each $T^2$ on the left and right boundary. On the left boundary $x = -M$, the small Wilson line operator for the semion along the plaquette $p$ is given by $A_{v_1}^2 B_p$, where the location of the vertex labeled by 1 relative to the plaquette $p$ can be found in Fig. 11. Multiplying this plaquette operator on a closed disk gives the extended Wilson line for the semion. Let us write the extended Wilson line operator in $y, z$ direction on the left boundary as $W_{y;l}, W_{z,l}$ respectively. Reflecting the fusion rule and mutual braiding of the semion, in the ground state Hilbert space we have

$$W_{y;l}^2 = W_{z;l}^2 = 1 \,, \qquad W_{y;l} W_{z;l} = -W_{z;l} W_{y;l} \,, \tag{C.5}$$

where the first equation uses that $C_e = 1$ for the ground states. The operators $\{W_{z,l}, W_{z,r}\}$ hence behave as the Pauli $x, z$ operators whose representation spans the 2d Hilbert space, so the left boundary carries the two-fold ground state degeneracy. The similar logic also holds for the right boundary, so the four-fold ground state degeneracy is totally carried by the $U(1)_2$ topological order on the left and right boundary. This implies that the reflection plane does not support the topological order.

## D  Property of the Grassmann integral

Here we review a quadratic property of the Grassmann integral on a square lattice introduced in the main text,

$$\sigma(\alpha) = \prod_e (d\theta_e d\theta'_e)^{\alpha_e} \times \prod_{p=(0123)} (\theta_{01}^{\alpha_{01}} \theta_{12}^{\alpha_{12}} \theta_{23}^{'\alpha_{23}} \theta_{30}^{'\alpha_{30}}), \tag{D.1}$$

with $\alpha \in Z^1(\Sigma, \mathbb{Z}_2)$ with $\Sigma$ a torus supporting a square lattice. We will show the quadratic property

$$\sigma(\alpha)\sigma(\beta) = \sigma(\alpha + \beta)(-1)^{\int_\Sigma \alpha \cup \beta}, \tag{D.2}$$

with $\alpha, \beta \in Z^1(\Sigma, \mathbb{Z}_2)$. The quadratic property of the Grassmann integral on the square lattice was derived in [46]. This can be shown by reordering the Grassmann variables in each term of $\sigma(\alpha + \beta)$. Firstly, the term on each plaquette is reordered as

$$\theta_{01}^{\alpha(01)+\beta(01)} \theta_{12}^{\alpha(12)+\beta(12)} \theta_{23}^{\alpha(23)+\beta(23)} \theta_{30}^{\alpha(30)+\beta(30)} =$$
$$= (-1)^{\beta(23)\alpha(30)+\beta(12)(\alpha(23)+\alpha(30))+\beta(01)(\alpha(12)+\alpha(23)+\alpha(30))}$$
$$\times \theta_{01}^{\alpha(01)} \theta_{12}^{\alpha(12)} \theta_{23}^{\alpha(23)} \theta_{30}^{\alpha(30)} \theta_{01}^{\beta(01)} \theta_{12}^{\beta(12)} \theta_{23}^{\beta(23)} \theta_{30}^{\beta(30)}$$
$$= (-1)^{\beta(23)\alpha(30)+\beta(12)(\alpha(01)+\alpha(12))+\beta(01)\alpha(01)} \theta_{01}^{\alpha(01)} \theta_{12}^{\alpha(12)} \theta_{23}^{\alpha(23)} \theta_{30}^{\alpha(30)} \theta_{01}^{\beta(01)} \theta_{12}^{\beta(12)} \theta_{23}^{\beta(23)} \theta_{30}^{\beta(30)}$$
$$= (-1)^{\alpha \cup \beta(0123)+\beta(12)\alpha(12)+\beta(01)\alpha(01)} \theta_{01}^{\alpha(01)} \theta_{12}^{\alpha(12)} \theta_{23}^{\alpha(23)} \theta_{30}^{\alpha(30)} \theta_{01}^{\beta(01)} \theta_{12}^{\beta(12)} \theta_{23}^{\beta(23)} \theta_{30}^{\beta(30)}, \tag{D.3}$$

where cup product $\alpha \cup \beta$ on a single plaquette $(0123)$ is evaluated as $\alpha(01)\beta(12) + \alpha(30)\beta(23)$. Then, the integrand on each edge is reordered as

$$d\theta_e^{\alpha(e)+\beta(e)} d\theta_e^{\alpha(e)+\beta(e)} = (-1)^{\beta(e)\alpha(e)} d\theta_e^{\alpha(e)} d\theta_e^{\alpha(e)} d\theta_e^{\beta(e)} d\theta_e^{\beta(e)}. \tag{D.4}$$

Combining the above two effects of reordering the Grassmann variables, one can see that the sign factor in the form of $(-1)^{\beta(e)\alpha(e)}$ on each edge cancels out. Hence, the reordering effect is solely expressed in terms of cup product after all, which shows Eq. (D.2).

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
