# Peer review of "Fermionic defects of topological phases and logical gates"

_SciPost Physics, doi:SciPost Phys. 15, 028 (2023)_

## Round 1 · Author Response

We are thankful to the pertinent comments and their positive evaluations from the three referees. Below let us reply to each of the suggestions raised by the referees.

Report 1: - The manuscript discusses defects that are not quite topological but depend on the spin structure. If Z2f is gauged on the defects, they should give completely topological defects, can the author comment on these defects? Reply: Thank you for suggesting to consider the defect obtained by gauging Z2f symmetry of the fermionic defects, which clarifies the relation to topological defects of the bosonic phase. We added their descriptions in the newly created Sec.II B 3, and found that the resulting defects are condensation defects of the bosonic topological phases.

Report 1: - Although the bulk system is supposed to be bosonic with local fermion only on the defects, some discussions in the manuscript e.g. p14 involve local fermions also in the bulk. Is the conclusion the same without bulk local fermion? Reply: Thank you for this pertinent comment, which was helpful for improving the quality of the manuscript. We added a discussion of the case where the local fermion is also introduced in the bulk, in the newly created Sec.II E. We found that the invertible defects of the (2+1)D bosonic phase stacked with an atomic insulator is again described by fusion of the gauged Gu-Wen SPT defect and the symmetry defect of the bosonic phase. So, the description of the invertible defects in the presence of the bulk local fermion is exactly same as the case without the bulk local fermion.

Report 3: - It seems invertibility for the defects produced by condensing a fermion along a line with and without physical fermions is different. In the former case I think the defect is invertible, in the latter case it is said in the paper that it is not. I guess this is due to differences in condensing the emergent fermion with or without physical fermions. Could the author clarify this? Reply: Thank you so much for this insightful comment. Indeed, the property of the fusion rule depends on whether the condensation defect involves a physical fermion or not. We explained this in the newly created Sec.II B 3, where the condensation defect without a physical fermion is obtained by gauging Z2f symmetry of that with a physical fermion.

We also modified the typos raised by Referee 3 and made minor editions to improve the readability. We believe that the above editions address all the suggestions, and now the manuscript is suitable for the publication to SciPost Physics.

---

## Round 1 · List of Changes

1. We added Sec.II B 3, which discusses the defects obtained by gauging Z2f fermion parity symmetry of the fermionic defects. We point out that the resulting defects are condensation defects of the anyons without involving the physical fermion, and clarified the role of the physical fermion in the gauged Gu-Wen SPT defect.
2. We added Sec.II E, which discusses the property of the topological defect of the fermionic theory obtained by stacking the (2+1)D bosonic topological phase with an atomic insulator. This corresponds to the emergent symmetry for the logical gate discussed in Sec.IV.
3. We modified the typos pointed out by one of the referees, and made minor revisions to improve the readability.

You are currently on this page

Resubmission scipost_202304_00020v1 on 23 April 2023

---

## Editorial Decision

published